# Designing Accurate Moment Tensor Potentials for Phonon-Related Properties of Crystalline Polymers

**DOI:** 10.3390/molecules29163724

**Published:** 2024-08-06

**Authors:** Lukas Reicht, Lukas Legenstein, Sandro Wieser, Egbert Zojer

**Affiliations:** 1Institute of Solid State Physics, NAWI Graz, Graz University of Technology, 8010 Graz, Austria; lukas.reicht@tugraz.at (L.R.); lukas.legenstein@tugraz.at (L.L.); sandro.wieser@tuwien.ac.at (S.W.); 2Institute of Materials Chemistry, TU Wien, 1060 Vienna, Austria

**Keywords:** machine-learned potential, moment tensor potential, active learning, molecular dynamics, phonon calculation, thermal conductivity, force field, polyethylene, polythiophene, P3HT

## Abstract

The phonon-related properties of crystalline polymers are highly relevant for various applications. Their simulation is, however, particularly challenging, as the systems that need to be modeled are often too extended to be treated by ab initio methods, while classical force fields are too inaccurate. Machine-learned potentials parametrized against material-specific ab initio data hold the promise of being extremely accurate and also highly efficient. Still, for their successful application, protocols for their parametrization need to be established to ensure an optimal performance, and the resulting potentials need to be thoroughly benchmarked. These tasks are tackled in the current manuscript, where we devise a protocol for parametrizing moment tensor potentials (MTPs) to describe the structural properties, phonon band structures, elastic constants, and forces in molecular dynamics simulations for three prototypical crystalline polymers: polyethylene (PE), polythiophene (PT), and poly-3-hexylthiophene (P3HT). For PE, the thermal conductivity and thermal expansion are also simulated and compared to experiments. A central element of the approach is to choose training data in view of the considered use case of the MTPs. This not only yields a massive speedup for complex calculations while essentially maintaining DFT accuracy, but also enables the reliable simulation of properties that, so far, have been entirely out of reach.

## 1. Introduction

Crystalline polymers have a peculiar structure: they are characterized by strong covalent bonds along their polymer chains and weak van der Waals (vdW) bonds in directions perpendicular to them. This leads to highly anisotropic characteristics, especially for phonon-related properties and heat transport, which are in the focus of our interest. A relevant reference material in this context is polyethylene (PE), which is also the structurally simplest polymer. In its common, amorphous form, PE is a thermal insulator with a very low thermal conductivity, *κ*, from around 0.3 to 0.5 Wm^−1^K^−1^ [1,2]. When its crystallinity is increased, its thermal conductivity increases dramatically. In fact, in (partly) crystalline PE thin films, thermal conductivities ranging from 22.5 Wm^−1^K^−1^ [3] and ~40 Wm^−1^K^−1^ [4] to even 62 Wm^−1^K^−1^ [5] and 65 Wm^−1^K^−1^ [6] have been measured, with the actual values depending on the quality of the samples. The highest thermal conductivity values have been observed in stretched fibers. For these, *κ* reaches values between 90 Wm^−1^K^−1^ [7] and 104 Wm^−1^K^−1^ [8].

To better understand the underlying mechanisms of thermal conduction and to know its theoretical limits, a reliable simulation of *κ* for a perfectly ordered single crystal is extremely helpful. To achieve this, a reliable simulation of harmonic and anharmonic phonon properties is required. The knowledge of phonon characteristics also provides insight into vibrational spectra, elastic material properties, and temperature-related quantities like thermal expansion. The precondition for correctly describing phonon properties is an accurate simulation of the forces acting on the polymer constituents upon displacing specific atoms. A complication in this context is that, when calculating anharmonic phonon properties, (tens of) thousands of individual force calculations need to be performed, especially when dealing with extended, complex systems containing many atoms in the considered supercells. Provided that such force calculations are possible at a sufficiently high level of accuracy, the material’s thermal conductivity can, for example, be calculated via the Boltzmann transport equation (BTE) by simulating phonon heat capacities, group velocities, and lifetimes [9]. Alternatively, with high-quality forces at hand, the heat transport properties can be calculated from particle trajectories, employing approaches such as non-equilibrium molecular dynamics (NEMD) [10], the Green–Kubo method [11,12], or by performing approach-to-equilibrium molecular dynamics (AEMD) [13] simulations. The number of required simulations is, however, even larger in such molecular dynamics (MD)-based simulations.

In either case, the repeated calculation of forces on atoms becomes so computationally costly that it is often infeasible when employing ab initio methods like density functional theory (DFT). Therefore, the community often resorts to empirical force fields, which can, however, be very inaccurate, especially when the phonons of interest describe intra-molecular vibrations. This can, for example, be concluded from the large spread in the thermal conductivities along polyethylene chains calculated with such “off-the-shelf” empirical force fields. Depending on the nature of the force field, the thermal conductivity along PE chains has been predicted to amount to ~50 Wm^−1^K^−1^ [14] (using the COMPASS force field), ~45 Wm^−1^K^−1^ [15] (employing the AIREBO potential) or 310 ± 190 Wm^−1^K^−1^ [16] (with the REBO potential). Notably, the former values are even significantly below the highest measured ones, although in experiments, the measured thermal conductivity is inevitably reduced by sample imperfections, which extrinsically enhances phonon scattering. Some of us have also recently shown that empirical force fields like COMPASS and GAFF fail at accurately reproducing the phonon band structure of molecular crystals like naphthalene, especially in regions in which intra-molecular vibrations dominate [17]. This is in sharp contrast to DFT (in particular PBE with the Grimme D3 or the many-body dispersion (MBD) correction [18,19,20]), which yields phonon band structures [17] and low-frequency Raman spectra [21] in excellent agreement with experiments. Thus, overall, van-der-Waals-corrected DFT accurately describes phonon-related properties, but is slow; conversely, transferrable, empirical force fields are fast, but often suffer from inaccuracies.

Recently developed machine-learned (force field) potentials (MLPs) promise to resolve this dilemma, especially when they are parametrized specifically for the material in question [22,23,24]. The latter can be achieved by training them on accurate DFT data. The type of MLPs in the focus of this paper are moment tensor potentials (MTPs), as implemented in the Machine Learning Interatomic Potentials (MLIP) code [25,26]. Their conceptual structure will be briefly discussed below. Since, to the best of our knowledge, MTPs have not been systematically used for polymers, we focus here on devising optimized strategies for their parametrization and benchmarking. This serves as a crucial prerequisite for using them to systematically understand the heat transport and related phenomena in polymers in follow-up studies.

To benchmark the obtained potentials, we will compare a range of observables calculated with the MTPs to the results obtained when using the same DFT approach that is used in the MTPs’ parametrization. When experimental data for certain properties are available as well, these will also be considered in the benchmarking. In the latter case, however, one has to keep in mind that a comparison to experiments not only tests the quality of the parametrization of the MTPs, but also the suitability of the underlying ab initio method (e.g., DFT).

In the context of thermal and phonon-related properties, MTPs have been shown to correctly predict phonon band structures and thermal conductivities of simple crystals (diamond, silicon, InAs, and BAs) and 2D materials [24]. Also in another study, MTPs have correctly predicted the phonons of 2D materials and the MTP-calculated thermal expansion of graphene agreed reasonably well with the DFT results [27]. Moreover, the MTP-derived phonon band structures of BC_2_N monolayers [28], diamond [23], and BAs [23] agree very well with the corresponding DFT results. Moreover, in recent studies, some of us showed that MTPs also provide an exceptionally accurate description of phonon-related properties for metal–organic frameworks (MOFs) [29] and, when suitably parametrized, can even reproduce the spin-dependent vibrational properties of MOFs [30]. Very recently, MLPs have been used to accelerate the thermal conductivity calculations of 103 inorganic materials [31]. However, these results cannot be directly transferred to crystalline polymers, as, due to their extreme anisotropy, a successfully parametrized MTP must be able to correctly describe particularly strong covalent interactions along the polymer chains and very weak van der Waals and multipole interactions perpendicular to them, potentially in conjunction with highly flexible side chains attached to the polymer backbones. In the present study, we will discuss under which circumstances MTPs can live up to this challenge.

In particular, we will focus on developing strategies for the parametrization of accurate, system-specific MTPs for various use cases. For this, different sets of reference structures for the parametrization are calculated using dispersion-corrected density functional theory, which allows for simultaneously describing covalent, electrostatic, and van der Waals interactions (for details see Section 3). On the one hand, we will benchmark the performance of the obtained MTPs for predicting unit cells, the elastic stiffness tensor, as well as harmonic and anharmonic phonon properties. All these quantities can be calculated in a quasi-static manner considering comparably small atomic displacements. On the other hand, we will also test the accuracy of the suitably parametrized MTPs for performing molecular dynamics simulations at room temperature, at which much larger atomic displacements occur. In this context, we will also assess to what extent it can be beneficial to parametrize distinct MTPs for such complementary use cases.

Structures of Polyethylene, Polythiophene, and Poly(3-hexylthiophene-2,5-diyl) (P3HT)

In the present study, we will investigate three polymers with complementary properties: polyethylene (PE), polythiophene (PT), and poly-3-hexylthiophene (P3HT), which are shown in Figure 1. Throughout the manuscript, these polymers are considered in their crystalline form, rather than in their amorphous state or as a melt. Therefore, in the simulations, periodic boundary conditions are used, which have the practical consequence of the polymer chains not having a beginning or an end. The unit cells periodically repeated in all directions of space are indicated by the black rectangles in Figure 1. PE is characterized by fully saturated single bonds along its chain. This results in a certain degree of flexibility of the polymer backbone. Moreover, it has a rather simple structure, with only 12 atoms in the unit cell, which also allows us to perform properly converged DFT calculations of phonon lifetimes, at least when restricting the description to three-phonon scattering processes (i.e., third-order force constants). Relying on state-of-the-art computational resources, this is no longer feasible for PT (28 atoms in the unit cell) and even less so for P3HT (100 atoms in the unit cell). In that sense, PT and P3HT serve as examples for what becomes possible when using machine-learned potentials. Here, PT is a classical reference system for a semiconducting polymer with a rigid, π-conjugated backbone. P3HT combines the features of PT and PE, as it consists of a rigid PT backbone surrounded by flexible hexyl sidechains that make it soluble in common organic solvents. From a practical point of view, P3HT is also one of the most applied conjugated polymers, as, for a long time, it served as a classical benchmark material for polymer-based organic thin-film transistors [32,33,34,35].

PE is typically observed as one of two polymorphs, being either orthorhombic or monoclinic [36]. The monoclinic polymorph is metastable and reverts back to the orthorhombic form when subject to temperatures in excess of 60–70 °C [36]. Thus, in the following, we will consider only the orthorhombic polymorph of PE, where, as a starting geometry for our geometry optimizations, we picked the known structure from Huan et al. [37]. To the best of our knowledge, the crystal structure of PT has not been unambiguously determined yet. Therefore, we constructed it assuming an orthorhombic unit cell with two chains, in accordance with previous studies [38,39]. The structure was then relaxed with DFT. More details on how the structure was created can be found in Appendix A of the Appendix A. For the starting structure of P3HT, we relied on the work of Zhugayevych et al., which deals with a theory-based identification of the lowest-energy polymorphs [40]. In that work, multiple nearly isoenergetic polymorphs were found, whose energies at room temperature differed by less than *k_B_T*. We opted for the lowest-energy polymorph labelled “CAaac” in the paper [40], whose structure and monoclinic unit cell were then relaxed with DFT. The structures were aligned with the chain directions parallel to the Cartesian *z*-direction**,** which is parallel to the lattice vector ***c***, in accordance with the literature [36,41]. In addition to the real space structures in Figure 1, Figure 2 illustrates the first Brillouin zones of PE and P3HT. The first Brillouin zone of PT is not shown, as it is analogous to that of PE.

**Figure 1 molecules-29-03724-f001:**
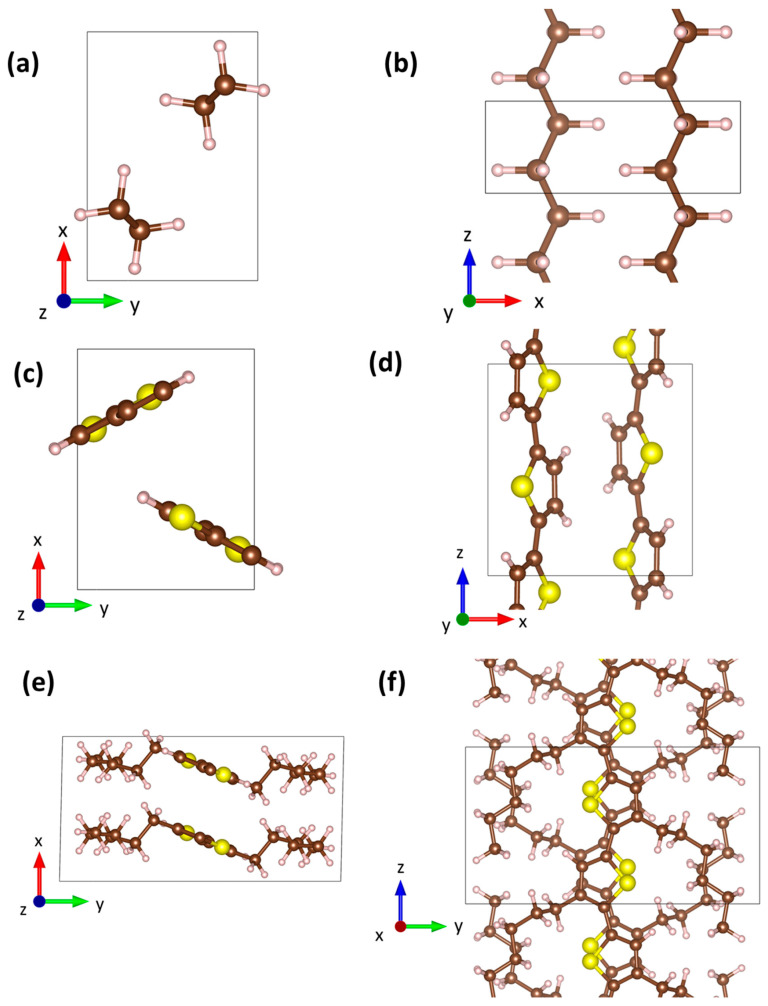
Optimized real space structures of orthorhombic polyethylene (PE) (**a**,**b**), of polythiophene (PT) (**c**,**d**), and of poly(3-hexyl-thiophene (P3HT) (**e**,**f**) with the viewing directions along the direction of the polymer chains (**a**,**c**,**e**) and perpendicular to them (**b**,**d**,**f**). The tripods indicate the cartesian directions *x*, *y*, and *z*. (Color code—brown: carbon; white: hydrogen; yellow: sulfur; plots produced using VESTA [42]).

For polyethylene (PE), all of the quantities considered in this work are calculated, because the corresponding DFT calculations are feasible and/or experimental data exist. For polythiophene (PT) and P3HT, selected quantities like the thermal conductivity and thermal expansion will not be benchmarked, as for these, no suitable reference data exist, and their calculation with DFT is beyond current computational possibilities.

## 2. Results and Discussion

In the following, we will first describe the details of the used MTPs together with the strategy used for their parametrization. Here, we will address the question of how the parametrization strategy should be adapted to the intended use case of the MTP. Subsequently, we will benchmark the performance of the “best” MTPs for predicting structural parameters, phonon bands, elastic constants, and anharmonic properties (like phonon lifetimes and thermal expansion). Finally, their performance in MD simulations and the achievable speedup of the computations will be assessed.

### 2.1. Generating Moment Tensor Potentials

#### 2.1.1. The Tested Moment Tensor Potentials (MTPs) and Their Training Process

In the following, a general overview of the employed computational approach will be given, which is necessary for assessing the results discussed below. The more technical details will be compiled at the end of the manuscript in Section 3. As mentioned in Section 1, the machine-learned potentials of choice for the present study are moment tensor potentials (MTPs), as implemented in the MLIP code (Version 2) and as described in detail in Refs. [25,26]. In short, in MTPs, the local environments around atoms are identified within spheres with a chosen radius. For all systems discussed below, this radius is set to the default value of 5 Å. We typically find a comparably weak dependence of the performance of the MTPs on this parameter, with the best results typically obtained around the default value. The energy contribution of an atom is then determined by the neighbors within its environment. Summing the contributions of all atoms yields the total energy of the system. The atomic contributions due to the local environments are expanded in terms of basis functions, which are built from moment tensor descriptors. These consist of a radial part represented by a series of polynomials up to a given maximum order (defined by the “radial basis set size” parameter), and an angular part, which is represented by a tensor up to a certain rank. The multitude of combinations of moment tensors included in the potential is defined by a “level” parameter. Increasing the level and the radial basis size increases the number of trainable parameters, which typically increases the accuracy of the MTP, but at the same time, reduces its speed. According to our experience, increasing the level has a stronger impact than increasing the radial basis size, both in terms of speed and accuracy. Therefore, throughout this work, we kept the radial basis size fixed to 10. The influence of the level is investigated by setting it to 18, 22, and 26 (see below).

In this context, it should be mentioned that, for the quantities of interest in the present study, even the highest-level MTPs are many orders of magnitude faster than any DFT simulation. Therefore, whether employing a computationally less efficient MTP could become a problem depends on the situation. This is best discussed in the context of specific use cases: one set of target applications of the MTPs discussed in this work is the simulation of harmonic and anharmonic force constants to obtain phonon band structures and phonon lifetimes. In the case of PE, only 12 force calculations on structures comprising a single, displaced atom are needed when calculating phonon band structures. In contrast, around 8000 force calculations on structures with pairs of displaced atoms are required for determining the phonon lifetimes from third-order force constants (for details see Section 3). The latter constitutes a considerable computational workload when performing the simulations with DFT. In contrast, such calculations are straightforward on any PC, even for the most complex MTPs considered here. Therefore, for such calculations, the speed of the MTP is not really an issue, and very high levels (like level 26) can be chosen. Conversely, when performing molecular dynamics (MD) simulations (here, for evaluating the thermal expansion of PE), the computational demand is usually significantly higher. This is particularly true when studying thermal transport with NEMD and AEMD approaches, where forces for systems comprising tens of thousands of atoms need to be calculated many millions of times. Thus, for MD simulations, it will typically be advisable to use a lower level, like 22, 18, or even 16, which makes the form of the MTPs mathematically less complex with fewer adjustable parameters. This results in a significant speedup of the MTPs (see Section 2.9, Appendix A, and especially Appendix A, as well as Ref. [29]).

Another relevant aspect of MTPs is that, by default, they treat all atoms of one chemical element equally, employing the same set of parameters. For PE, this is perfectly fine, as all carbons have identical chemical environments. However, if one considers more complex polymers, carbon atoms can appear in significantly different configurations (i.e., hybridizations, chemical environments, etc.). In such cases, it is beneficial to treat them as different atom types. Therefore, for PT, we treat carbon atoms as two atom types, distinguishing between carbon atoms bonded to two other carbon atoms and one hydrogen and carbon atoms bonded to two carbons and one sulfur. In P3HT, the chemical environments of the carbons are even more diverse, as illustrated in Figure 3. A major aspect is that now one is dealing with sp^2^-hybridized carbon atoms in the backbone and sp^3^-hybridized ones in the side chains. Describing the differently hybridized carbons by different atom types turns out to be crucial for the accuracy of the MTPs. Additionally, we will also assess to what extent further differentiation between carbon atoms in non-identical chemical environments would be useful. These different atom types are illustrated by the different shadings of red and blue in Figure 3 (see also figure caption). In terms of speed, distinguishing between atom types significantly increases the required computational effort to parametrize the MTPs due to the larger number of parameters that need to be fitted (a quantitative assessment will be provided below). However, a larger number of atom types has no significant impact on the runtime of simulations when using an already parametrized MTP.

Training MTPs means adjusting their variable parameters, such that a cost function is minimized. In MLIP, the cost function comprises the weighted square deviations between the energies, stresses, and forces calculated by the respective MTP and a reference methodology for certain training structures. In our case, that reference methodology is dispersion-corrected DFT employing periodic boundary conditions. In short, the PBE functional [43] is combined with Grimme’s D3 correction [18,19] (with Becke–Johnson damping [20]) to account for long-range van der Waals interactions. For these simulations, converged basis sets and *k*-point grids are employed and the Vienna Ab initio Simulation Package (VASP) [44] is used (further technical details are provided in Section 3). The generation of the training structures will be described in Section 2.1.2. The different contributions to the cost function are scaled, such that one can give more weight to either energies, stresses, or forces. Here, we use the default weights of 1 (eV)^−1^, 0.01 Å(eV)^−1^, and 0.001 (GPa)^−1^ for energies, forces, and stresses, respectively. Training ends when the cost function drops by less than a factor of 10^−3^ over the previous 50 iterations.

The random initialization of the fit parameters at the start of the training results in a stochastic nature of the obtained MTPs [26]. Therefore, it is useful to train ensembles of differently initialized MTPs. We typically train five MTPs for any system and level of MTP complexity. These five MTPs are used to calculate specific quantities of interest (i.e., they are benchmarked on a validation set [45]), and then the “best” MTP is picked for further use. For MTPs meant for primarily modeling phonon properties, the selection criterion could, for example, be their ability to reproduce Γ-point frequencies or phonons in the whole Brillouin zone, where the latter is adopted here. Thus, phonon frequencies homogeneously sampled over the entire Brillouin zone are used to calculate the root mean square deviation (RMSD) up to 12.5 THz for choosing the “best” MTP that is meant for phonon calculations. The reason for restricting the frequency range will be explained in Section 2.2, where also the deviations between different parametrization runs will be discussed. A summary of various possible quality metrics for the MTPs is provided in Appendix A.

Conversely, for ranking the MTPs meant for MD simulations, a possible criterion that is also chosen here is their performance when describing the forces acting on atoms in a set of validation structures not used in the parametrization (see Section 3 for details on the generation of the validation structures). As an alternative to picking the “best” MTP, one can calculate the quantities of interest with all five MTPs, and then take the mean and standard deviation of the different predictions. This approach was used by the MLIP developers in their original paper [26] and it has the advantage of obtaining an uncertainty estimate by the means of the standard deviation, which can, for example, be compared to the difference between the MTP- and DFT-calculated quantities. Thus, for selected quantities, in the following, mean values will also be specified.

#### 2.1.2. Generating the Training Data for the MTP Parametrization

When generating reference data for the parametrization of MTPs, it is crucial to perform an efficient sampling of the configuration space. For that, different strategies have been employed in the past. To benchmark these approaches in the context of heat transport modeling, Choi et al. [46] generated data for training neural network interatomic potential (NNIP) by ab initio molecular dynamics, by random displacements of atoms, and by generating displacements as a superposition of phonon eigenmodes. They concluded that the best-performing NNIPs were those generated based on the ab initio molecular-dynamics-derived training data. Here, we employ a similar, yet significantly more computationally efficient approach: in short, for all our systems, we perform molecular dynamics simulations within VASP relying on DFT accelerated by on-the-fly parametrized kernel-based force fields [47]. In this approach, VASP uses a number of initial ab initio MD steps to generate a VASP machine-learned potential (VMLP), which is used to predict further MD steps. Depending on whether the Bayesian error estimation lies below or above a threshold, that predicted step is retained or a step using DFT-calculated forces is performed. The threshold of the error can be automatically determined or user-defined (as further discussed in Appendix A). Any new DFT calculation then becomes part of the pool of DFT reference data that is available for the training of the VMLP. This process continues, and the pool of reference data grows over time until enough training data have been collected. In our cases, such an active learning run typically occurs for 10,000 to 100,000 MD steps, comprising several hundred DFT simulations (with the actual numbers for each parametrization listed below). Due to the nature and parametrization of the VMLP, this strategy is particularly computationally efficient (compared, e.g., to MLIP’s internal active learning approach), as argued in more detail in [29].

The DFT-calculated energies, forces, and stresses from the VASP on-the-fly learning process are then used as training data for fitting the MTPs. This two-step training approach has, for example, been implemented by the VASP group for modeling interfaces [48]. A detailed comparison of the performances of VMLPs and MTPs in terms of speed and accuracy can be found in [29], suggesting similar accuracies of both potential families and, at the time of the study, somewhat faster simulations with MTPs. Most importantly, the MTPs can be straightforwardly interfaced with popular molecular dynamics codes like LAMMPS (Large-scale Atomic/Molecular Massively Parallel Simulator) [49] using the LAMMPS–MLIP interface. This boosts the versatility of MTPs and makes them particularly suited for dealing with all use cases in the focus of the present study on an equal footing. Therefore, in the following, we will exclusively discuss the performance of the MTPs.

A central question for any MD-based generation of training data is at what temperature the MD simulations should be performed, or whether employing a temperature ramp during the run would be more appropriate. We opt for the latter in line with most previous active-learning endeavors employing VASP [29,48,50]. But also, in that case, a maximum temperature for the heat ramp needs to be set, where it again appears advisable to consider the envisioned use case of the MTPs: in single-point calculations geared at obtaining harmonic phonon properties, all atoms except for one are at their equilibrium positions, while the one atom is typically displaced by 0.01 Å (a value that we checked for convergence, and which is known to provide well-converged results for organic materials [17]). Similarly, when calculating phonon lifetimes via finite displacements of pairs of atoms, the considered atomic displacements are very small. For example, the default displacement value in phono3py [9] (which is also applied here; for convergence tests, see Appendix A) amounts to only 0.03 Å. As a consequence, in such calculations, the MTPs are required to accurately describe the very weak forces on atoms in structures that are particularly close to equilibrium. Consequently, for calculating harmonic and anharmonic phonon properties, training data at small displacements (i.e., at low temperatures) ought to be sufficient or even preferable. Thus, for calculating phonon properties, we test the impact of generating training data only up to 100 K. In contrast, when using MTPs for performing MD simulations around room temperature, much larger displacements are encountered. For example, for the polymers considered here, the average atomic displacements in MD runs amount to roughly ~0.13 Å at 15 K and ~0.6 Å at 300 K, as described in Appendix A. Thus, when performing MD simulations, especially at room temperature, the MTP must accurately describe forces at comparably larger displacements. This suggests that the use of training data at elevated temperatures—possibly even significantly above the envisioned target temperature—would be advisable. In fact, including high-temperature reference data is a commonly applied strategy for improving the force fields eventually used for molecular dynamics runs [29,51]. Also, the VASP manual suggests overshooting the application temperature in the training by about 30% [51]. Thus, for the MD use case of the present study, training data are sampled up to a temperature of 500 K, which is significantly higher than the maximum temperature of 300 K used when simulating thermal expansion. Notably, at such high temperatures, polymers like PE or P3HT would be melts in their equilibrium configurations. However, the simulation times applied are much too short for the polymers to melt, such that, in the simulations, they remain in their crystalline structure. Moreover, the high-temperature simulations are exclusively performed for sampling the configuration space for parametrizing machine-learned potentials.

### 2.2. Impact of the Choice of the Reference Data, the Level of the MTP, and the Number of Considered Atom Types

Prior to providing an extensive, quantitative comparison between MTP- and DFT-calculated observables, it is useful to assess the impacts of several factors influencing the parametrization process. This, in particular, applies to the reference data, the MTP level, and the number of considered atom types. Adjusting these settings provides handles for improving the accuracy of MTPs for the two use-cases outlined above, namely for calculating phonon-related properties (using small atomic displacements) and for performing MD runs at finite temperatures (associated with much larger atomic displacements). For assessing the various, differently parametrized potentials, two metrics are defined: to quantify the accuracy of MTPs in predicting phonons, the root mean square deviation (RMSD) between the MTP phonon frequencies and the DFT reference data, RMSD^phonon^, is calculated on a dense mesh in the whole Brillouin zone up to 12.5 THz throughout the paper (further details, e.g., on the mesh size are given in Section 3). The choice of a cutoff is motivated by the fact that low-frequency phonons are insofar more relevant, as their thermal occupation is higher, and they typically have higher group velocities and phonon lifetimes. This makes them the main contributors to the thermal conductivity. As shown in Appendix A, the exact value of the cutoff is not significant. As a second parameter complementary to RMSD^phonon^, a metric to assess the accuracy of the MTPs in MD simulations is defined. It is referred to as RMSD^MD^ and is calculated as the RMSD between the MTP-calculated forces and DFT forces for a validation set generated via a 300 K active learning run (see Section 3 for details on the generation of the validation set).

With these two metrics (RMSD^phonon^ and RMSD^MD^) in hand, three strategies for improving the accuracy of the MTP will be assessed. The first is optimizing the generation of the training data, where, for the reasons discussed in Section 2.1.2, the focus is on the impact of the temperature range over which the training data are generated. The “default” training sets for PE and PT originate from active learning MD runs from 15 K to 500 K in an *NPT* ensemble with 150 MD steps per Kelvin temperature increase. The “default” MTPs are then parametrized at level 22 with one carbon type for PE and two carbon types for PT. To test the impact of emphasizing training data at low temperatures, additionally smaller reference data sets are defined by including only the subsets of DFT-calculated structures obtained in temperature ranges from 15 K to 100 K, 15 K to 200 K, 15 K to 300 K, and 15 K to 400 K. For PE (PT), this yields five training data sets with 129 (173) configurations generated at temperatures ranging from 15 K to 100 K, 260 (260) configurations from 15 K to 200 K, 368 (344) configurations from 15 K to 300 K, 507 (456) configurations from 15 K to 400 K, and the original 550 (490) configurations from 15 K to 500 K. The resulting RMSD^phonon^ are then calculated for five independently parametrized MTPs for PE and PT employing a DFT-relaxed unit cell. The resulting deviations between the MTP and DFT results are shown in Figure 4a for PE (blue) and PT (orange).

As will be discussed in more detail below, the frequency deviations are generally very small. Nevertheless, the random initialization of the MTP fitting procedure leads to a non-negligible spread within each set of five MTPs parametrized with identical settings. The majority of the MTPs within each set result in rather similar and small values of RMSD^phonon^, with typically one or, at most, two outliers. The RMSD^phonon^ values for the best-performing MTPs are shown in Figure 4a by large, open symbols connected by dashed lines. The values obtained for all other MTPs are shown by small, solid symbols, with the values for the median MTPs connected by solid lines. In view of the outliers obtained in each temperature range, stressing the median values appears more useful than calculating the average values of RMSD^phonon^. The trends for both the median and the best-performing MTPs show that using only low-temperature training data clearly improves the RMSD^phonon^ for PE as well as for PT. For the best-performing MTPs, in the case of PE, the value of RMSD^phonon^ drops by more than a factor of two from 0.15 THz (for 15–500 K reference date) to 0.07 THz (for 15–100 K reference data). Similarly, for PT, it drops from 0.08 THz to 0.04 THz. This is remarkable insofar as the number of reference configurations used in the parametrization is reduced by a factor of more than two for the restricted temperature range. The improved accuracy arises from the fact that the forces in the phonon calculations are more similar to the forces encountered in the training data sampled at a low temperature than to those occurring during training at higher temperatures (as illustrated in more detail in Appendix A). To test whether the “low-temperature” MTPs can be further improved by increasing the number of reference configurations, we repeat the active learning run for PE between 15 K and 100 K, starting with the already obtained data from the run between 15 K and 100 K (to avoid adding very similar configurations). This increases the number of training configurations from 129 to 397. We then reparametrize five new MTPs on the expanded “low-temperature” reference configurations. The resulting RMSD^phonon^ remains virtually unchanged at 0.72 THz for the respective best-performing MTPs, while the median RMSD^phonon^ decreases somewhat from 0.10 THz to 0.08 THz (which appears to be within the statistical fluctuations). Also, fixing the unit cell size in the training (employing an *NVT* instead of an *NPT* ensemble) does not yield statistically relevant changes in RMSD^phonon^ (with hardly any change for the best-performing MTPs and some decrease for the median, see also Appendix A). A possible reason for this is the very small thermal expansion of PE below 100 K.

As a next step, the potentials, assessed in terms of their accuracy when evaluating the phonon properties (Figure 4a), are also tested in terms of their accuracy in predicting the forces for the structures encountered in MD runs at 300 K (Figure 4b). The values of RMSD^MD^ follow an opposite trend compared to the values of RMSD^phonon^: the larger the temperature range considered in the generation of reference configurations, the more accurate the force predictions become. This is not unexpected, considering that the MTPs trained on the data obtained from active learning runs between 15 K and 100 K have never encountered the large atomic displacements occurring in the validation structures at 300 K. In view of this, it might appear surprising that the performance of the “low-temperature” MTPs is still reasonable, but for MD simulations, at elevated temperatures, they fail in a different context: the simulations become unstable, resulting in a disintegration of the polymer chains. This is explicitly tested for the best-performing “low-temperature” MTP of PE in 800 ps *NPT* runs using supercells containing 3456 atoms for fixed temperatures set to values between 100 K and 340 K in 40 K steps. While the simulations are stable up to 220 K, the polymer chain disintegrates at all tested higher temperatures.

To assess the impact of the level on the MTPs’ accuracy, MTPs trained with data generated between 15 K and 100 K are parametrized at level 18, level 22, and level 26. Figure 4c,d clearly show that both RMSD^phonon^ and RMSD^MD^ decrease with increasing level. In fact, RMSD^phonon^ for the best-performing MTP of this series decreases by more than a factor of two between level 18 and level 26, and also, for RMSD^MD^, this decrease amounts to 33%. Moreover, while for the level 22 MTPs, one of the stochastically initialized MTPs is still an outlier, generating comparably large deviations, in the present test, this does not occur at level 26. This is consistent with our observation that higher MTP levels tend to decrease the stochastic spread of the MTPs. A higher MTP level, however, comes with higher computational costs both in the parametrization and in production calculations, as discussed in Section 2.1.1. As a rule of thumb, in our experience, increasing the level by two increases the computational demands by roughly a factor of 1.5, as illustrated in Appendix A of the Appendix A.

As a third strategy for increasing the accuracy of the MTPs (especially in complex materials), different atom types can be assigned to identical elements in different chemical environments. P3HT is the ideal test system for the following comparison, considering the significant number of chemically non-identical carbon atoms (see Figure 3). In Figure 5, the RMSD^phonon^ and RMSD^MD^ values for sets of level 22 MTPs trained with reference data between 15 K and 500 K are compared for different degrees of atom typing of the carbons of P3HT: this includes treating all carbons in P3HT equally, introducing three atom types (distinguishing between carbon in the side chains, carbon in the backbone bonded to sulfur, and carbon in the backbone not bonded to sulfur), and using six atom types (distinguishing between all chemically inequivalent carbons, as illustrated by the different shadings of red and blue in Figure 3). Splitting into three types gives a clear improvement compared to a single atom type for both RMSD^phonon^ and RMSD^MD^. Splitting into six types improves the RMSD^phonon^ and RMSD^MD^ of the best and median MTPs even further. Notably, the arithmetic mean RMSD^phonon^ actually does not improve, but this is the consequence of a statistical fluctuation caused by a particularly poorly performing outlier for the MTP with six atom types (see Appendix A). In passing, we note that, also for PT, a significant improvement in the MTP accuracy is observed, when the chemically inequivalent carbon atoms are treated as separate atom types (see Appendix A). Speed wise, as already discussed in Section 2.1.1, splitting into atom types does not decrease the speed of the MTPs, but it somewhat increases their training time (by ~60% for three atom types and by ~170% for six types, when using the computing hardware described in Appendix A). Thus, maximizing the number of atom types in a material appears advisable, as long as this does not have any adverse consequences on the stability of the MTPs. In fact, when using MTPs for P3HT in MD, the simulations tend to be more stable for three atom types for carbon compared to six.

For designing the MTPs used in the following benchmarking (and also in further calculations), all three discussed strategies can be combined to boost their overall accuracy. For example, for P3HT, when using training data generated between 15 K and 100 K, setting the MTP level to 26, and splitting the carbon atoms into six types, the RMSD^phonon^ for the best-performing MTP (out of two) amounts to 0.04 THz (i.e., 0.8 cm^−1^). This is remarkably low, considering that P3HT is a rather complex system with 100 atoms in the primitive unit cell. As the above settings (level 26, training between 15K and 100 K, and a maximum number of atom types for carbons) are ideal for phonon calculations, they are suggested for that task. Thus, MTPs parametrized using the said settings in the following will be referred to as MTP^phonon^. Furthermore, we also define a suggested MTP for MD simulations (abbreviated by MTP^MD^), which needs to be considerably faster and must also work at elevated temperatures. It is based on training data from 15 to 500 K, has a level of 22, and also maximally split atom types. Amongst all parametrized MTP^phonon^ for a specific material, the one with the smallest RMSD^phonon^ in the following will be referred to as the “best” MTP^phonon^. Likewise, the MTP^MD^ with the smallest RMSD^MD^ will be considered to be the “best” MTP^MD^. In the following, the predictive performance of these “best” MTPs will be assessed for a variety of physical observables.

### 2.3. Predicting Unit Cell Parameters

Phonon frequencies (and other phonon-related properties) strongly depend on the lattice parameters used in their calculation. Therefore, as a first assessment of the accuracy of the machine-learned potentials, static lattice parameters calculated with MTPs and DFT via minimizing the potential energy are compared. Where available, experimental values are also reported. The results can be found in Table 1.

For all materials, an excellent agreement between the lattice parameters obtained with DFT and those with the MTPs is observed. In the vdW bonding directions (lattice vectors ***a*** and ***b***), the deviations are around ~1%, while in the chain direction (lattice vector ***c***), the agreement is even better, with deviations below 0.1%. Generally, the MTPs^phonon^ perform slightly better than the MTPs^MD^, which is not surprising, considering that the former set of MTPs is designed for describing structures close to equilibrium. In Table 1, we also report an uncertainty estimate that we obtain by parametrizing five MTPs and calculating the standard deviations of the predicted lattice parameters. Interestingly, for MTP^MD^, this uncertainty estimate is typically of a comparable magnitude as the deviation of the mean values of the MTP calculations from the DFT results, whereas for MTP^phonon^, the uncertainty estimate is smaller.

Importantly, DFT and MTP relaxations not only yield equivalent lattice constants, but also yield the same symmetries for all materials. PE and PT have an orthogonal unit cell (all angles, including *α*, are 90°). P3HT has a monoclinic symmetry with *α*, the angle between the ***b*** and ***c*** lattice vector, being slightly below 90°. When comparing the MTP-simulated unit cell parameters to experiments, the overall agreement is still convincing. For most values, the deviations are, however, clearly larger than the deviations from the DFT structures. This is due to three reasons: (i) there is a non-negligible experimental error manifested in the deviations between different experiments that are sometimes even larger than the deviations in individual experiments from the DFT results, (ii) several of the experiments are performed at elevated temperatures, while, for the simulated values reported in Table 1, thermal expansion is not considered (compare Section 2.7), and (iii) even though PBE + D3 used in the DFT simulations is known to provide a very good description of the structures of molecular crystals, it is certainly not perfect [56,57], and one cannot expect a machine-learned potential to perform better than the ab initio methodology used for its parametrization. Overall, the MTPs are still capable of excellently reproducing the densities of the crystalline polymers.

### 2.4. Elastic Constants

The elastic constants of polymers are of practical and fundamental relevance, as illustrated, for example, by the exceptionally high measured values of ~200–300 GPa for the Young’s modulus of highly oriented PE in the chain direction [58,59,60]. Moreover, the elements of the elastic stiffness tensor can be regarded as phonon-related properties, since they are intimately coupled to the group velocities of acoustic phonons via the Christoffel equations [61]. For calculating the elastic constants using the MTPs, we applied two approaches: the “clamped ion method”, in which the atomic positions are not relaxed directly for strained unit cells. Instead, the effect of the relaxation is estimated via the Hessian matrix [62] to save computational time. This approach is also the one used internally in VASP for calculating elastic constants. Additionally, for the MTP-based calculations, we also employ the “relaxed ion method”, in which the atomic positions for strained unit cells are relaxed directly. For PE and PT, the resulting elastic constants, calculated with identical MTPs, are very similar, irrespective of the used method, but for P3HT, the two methods give rather different results for all elastic constants apart from *C_zz_* (see Appendix A). This discrepancy shows that there is an issue with the “clamped ion method” for P3HT. We attribute this to a numerical instability and difficulties in estimating the atomic relaxation via the Hessian matrix discussed in Appendix A of the Appendix A. As the “relaxed ion method”, which does not suffer from this effect, is beyond reach in the DFT calculations of P3HT, the following benchmark for P3HT will only be concerned with *C_zz_*. Correspondingly, Figure 6 compares all elements of the elasticity tensor calculated with DFT and with MTPs as a parity plot only for PE and PT. Overall the agreement between DFT and the MTPs is quite satisfactory, especially for MTP^phonon^, which provides a better agreement than MTP^MD^. On the one hand, we attribute this to the higher level of MTP^phonon^. On the other hand, the small unit cell changes in the training data for MTP^phonon^ are closer to the small unit cell changes of 1% that are applied in the calculations of the elastic constants.

To provide a more quantitative comparison, we focus on the *C_zz_* element, which is by far the largest element and corresponds to how much stress along the chain is required to deform the material in that direction. Naturally, as there are strong covalent bonds along the chain, *C_zz_* reaches large values of 328 GPa, 383 GPa, and 151 GPa for PE, PT, and P3HT when calculated with DFT (with the full elastic stiffness tensor given in Appendix A). To understand why *C_zz_* is 2.5 times larger in PT than in P3HT, one has to consider that, for the same number of polymer chains in the unit cell, the area of the x-y plane is 2.7 times larger in P3HT than in PT (see Figure 1c,d). Comparing the “best” MTP^phonon^ and DFT, *C_zz_* deviates by 1.6%, 0.8%, and 0.1% (for PE, PT, and P3HT), respectively. This can be regarded as an excellent agreement between the two approaches. The “best” MTP^MD^ performs somewhat worse (albeit still very good) with deviations of 3.1%, 2.4%, and 2.7% (for PE, PT, and P3HT, respectively). To put the agreement between MTPs and DFT into perspective, DFT calculations are also performed with the default VASP energy cutoff of 400 eV rather than the converged cutoff of 900 eV (PE) or 700 eV (PT and P3HT). These calculations with the low-energy cutoff yield deviations that are larger than the deviations between the MTPs and DFT calculations with converged cutoffs they were trained on (see Appendix A).

For PE, a comparison to a large number of experiments is also possible, where the list of experimental data in Table 2 is even truncated, omitting X-ray diffraction and Raman spectroscopy results from before 1980. A more complete list is provided in Ref. [63]. Experiments typically only report the elastic stiffness tensor elements *C_zz_* and show a significant spread, which could be due to imperfect PE crystals [64]. In fact, especially for *C_zz_*, the spread in experimental data is much larger than the deviations between the MTP and DFT results. Notably, however, if one attributes too small experimental values to crystal imperfections and, thus, considers the largest values to be closest to the ideal, intrinsic situation, the agreement of the experiments with both MTP and DFT data is indeed excellent. Regarding *C_xx_* and *C_yy_*, the agreement is clearly already worse for the DFT data.

For PT and P3HT, the literature is sparser than that for PE. In fact, for these materials, values varying by orders of magnitude have been reported for different samples, and the directions for which the elastic constants were determined were often not clear [65,66]. Thus, we shifted the discussion of these data to Appendix A of the Appendix A.

**Table 2 molecules-29-03724-t002:** Elastic constants *C_xx_*, *C_yy,_* and *C_zz_* (also called Young’s moduli) of PE calculated in this study and from the literature for temperatures *T* (“RT” denotes room temperature). They are calculated with five MTPs (each for MTP^phonon^ and MTP^MD^), for which the arithmetic mean and standard deviation are calculated. Corresponding values are denoted as “mean” in the table. Additionally, the results with the “best” MTPs are listed.

First Author	Year	Method	*T* [K]	$C_{xx}$ [GPa]	$C_{yy}$ [GPa]	$C_{zz}$ [GPa]
Theoretical						
This study	2024	DFT	0	12.2	11.4	328
This study	2024	MTP^phonon^, “best”	0	13.2	12.2	322
This study	2024	MTP^phonon^, mean	0	13.0 ± 0.6	12.4 ± 0.5	322 ± 1
This study	2024	MTP^MD^, “best”	0	13.9	13.8	318
This study	2024	MTP^MD^, mean	0	15.8 ± 2.1	12.8 ± 2.0	316 ± 4
Kurita [63]	2018	DFT	0	10.9	7.8	333
Experimental						
Matsuo [67]	1986	X-ray	293			213–229
Nakamae [58]	1991	X-ray	RT			235
			117			254
Kobayashi [68]	1983	Raman	RT			281
Tashiro [69]	1988	Raman	RT			260
Pietralla [59]	1997	Raman	RT			315
Holliday [60]	1971	neutron	298	6	6	329
Twisleton [70]	1982	neutron	76	9	8	326

### 2.5. Phonon Band Structure

As a first directly phonon-specific property, the MTP and DFT calculated phonon band structures are compared. In Figure 7, we show the results for the best-case scenario, i.e., using MTP^phonon^ for all three systems. This yields a truly amazing agreement between the MTP and DFT results, even for P3HT, despite its highly complex band structure. In this context, it should be noted that the deviations for P3HT in Figure 7c are amplified, as, for the sake of clarity, the plotted frequency range is only half that of the other polymers (with a plot of the full frequency in Appendix A). Importantly, the calculated band structures also agree very well with the experimental data on the dispersion of acoustic phonons in PE along the Γ → Z direction deduced from the overtones of Raman spectra, as discussed in Appendix A in the Appendix A. Another aspect that needs to be stressed for phonon band structures is that excellent agreement is observed not only along the chain direction (which is parallel to Γ → Z), but also for the directions in which inter-chain interactions are dominated by vdW and multipole interactions (parallel to Γ → X and Γ → Y). This illustrates that the MTPs used here are able to simultaneously mimic a highly anisotropic bonding situation that leads to large and small forces occurring in different directions.

In view of the amazing agreement between the DFT- and MTP-calculated phonon band structures, the questions arise of whether this is a direct consequence of choosing the “best” potentials for that task (the MTP^phonon^ potentials with the minimum RMSD^phonon^) and how the accuracy would change for the MTP with the median RMSD^phonon^. To quantify the agreement, again, the values of RMSD^phonon^ are used, which are particularly well suited for that task considering that they represent the deviations between the MTP- and DFT-calculated phonon frequencies for a homogeneously sampled Brillouin zone up to 12.5 THz. The corresponding values are summarized in Table 3 for various MTPs. For the MTP^phonon^ used to plot the band structures in Figure 7, they amount to 0.043 THz (1.43 cm^−1^), 0.029 THz (0.97 cm^−1^), and 0.036 THz (1.20 cm^−1^) for PE, PT, and P3HT, respectively, i.e., they are extremely small, even for the structurally highly complex P3HT crystal. This agreement only very mildly deteriorates for other MTP^phonon^ variants, such that, for the MTPs with the median RMSD^phonon^ values, the deviations increase by only ~0.01 THz (also see Appendix A in the Appendix A for the actual band structures).

When using MTP^MD^ variants, the agreement becomes clearly worse, with the values of RMSD^phonon^ increasing by roughly a factor of three (see Table 3). Notably, this increase is much larger than that seen between the “best” and the median potentials for all variants. Even worse, the “best” MTP^MD^ of PE produces two modes with imaginary frequencies at the Y-point (shown in Appendix A). This further supports the notion that, when using MTPs for calculating phonon properties, the “phonon” parametrization strategy defined in the last paragraph of Section 2.2 is advisable.

As a last note, concerning low-frequency phonon band structures, it should be mentioned that, for the above-discussed simulations, DFT-optimized unit cells were used in all cases in order to not mix aspects regarding the accuracy of calculating phonon bands with the accuracy of predicting unit cells. We also test the use of an MTP-optimized unit cell for calculating phonon bands with the “best” MTP^phonon^. Especially for PE and P3HT, this causes only a minor increase in RMSD^phonon^ to 0.053 THz and 0.046 THz, respectively (see also Appendix A). For PT, the increase in RMSD^phonon^ is larger (to a value of 0.094 THz, which is similar to the value obtained for the MTP^MD^ with the DFT unit cell). This is insofar not surprising, as for PT, the deviations between the DFT- and MTP^phonon^-optimized unit cell parameters are also larger (see Table 2). Still, even in this case, the changes in the band structure are rather minor, as shown in Appendix A.

For benchmarking phonons over the entire frequency range, plotting band structures is not ideal, as the resulting plots would become far too busy. Thus, Appendix A in the Appendix A compare the phonon densities of states for the “best” versions of MTP^phonon^. Again, the agreement is extraordinary. To illustrate that the excellent agreement between the phonon band structures calculated with DFT and classical force fields cannot be taken for granted, Figure 7d compares the DFT-generated bands with those obtained using a traditional, transferrable force field for PE. Here, we choose the AIREBO potential [71], which has been commonly used for PE, even as recently as 2020 [8,15,72]. While the AIREBO potential does reproduce the overall shape of the phonon band structure of PE, it severely (in some instances, even by a factor of two) overestimates the frequencies of all phonons. This also results in significantly too large group velocities, which makes this force field unsuitable for calculating, for example, thermal conductivities, which will be the topic of Section 2.6.

### 2.6. Calculating the Thermal Conductivity of PE Using the Boltzmann Transport Equation

After demonstrating that the MTPs provide an excellent performance when calculating harmonic phonon properties, the question arises whether this also applies to anharmonic quantities, like phonon lifetimes. These are, for example, relevant for thermal conductivities. For PE, (with considerable computational efforts) we were able to use DFT with converged numerical settings to calculate the phonon lifetimes from third-order force constants. These phonon lifetimes were then used to calculate the thermal conductivity via the Boltzmann transport equation (BTE). For more complex materials like PT and especially P3HT, performing such calculations is clearly beyond current computational possibilities. Thus, the benchmarking in this section focuses solely on PE. In this context, it also needs to be stressed that the present section deals with a comparison of DFT and MTP results, employing an approach that is consistent with what has been used in the literature [72,73,74]. There are several conceptual shortcomings of this approach that need to be considered for a quantitatively accurate description of thermal conductivity. As they have no direct impact on the benchmarking performed here, they will only be discussed briefly at the end of this section.

The linearized form of the BTE can be solved directly using, for example, phono3py [75]. We refer to this as the “full BTE” solution. Another approach is to solve the BTE within the relaxation time approximation (RTA) [75]. This requires less computational effort and is commonly conducted in the literature, as it typically has comparably little impact on the results for materials with low thermal conductivities [76]. For PE along the chain direction, where thermal conductivities are large, the RTA has been shown to underestimate the thermal conductivity by ~20% as compared to the full BTE [72,73]. Nevertheless, we will also report results obtained within the relaxation time approximation, as this allows for a phonon-resolved analysis, which is beneficial for benchmarking purposes. The BTE expression for the thermal conductivity within the RTA is
(1)$$\kappa_{RTA}=\frac{1}{N_{q}V_{c}}\sum_{\lambda }C_{\lambda }v_{\lambda }⊗v_{\lambda }\;\tau_{\lambda }.$$

Here, $\kappa_{RTA}$ is the thermal conductivity tensor, $N_{q}$ is the number of *q*-points in the sampling of reciprocal space, $V_{c}$ is the volume of the unit cell, $C_{\lambda }$ is the heat capacity associated with the phonon mode λ, $v_{\lambda }$ is the group velocity vector of that mode, and $\tau_{\lambda }$ is its lifetime. The composite index, λ, contains the wave vector ***q*** and the band index *n* of each phonon, and the sum is formed over all phonon modes, λ. The mode heat capacities and the group velocities are harmonic phonon properties that can be obtained from the phonon bands, for which we already know that the MTP^phonon^ results perfectly match the DFT data. Phonon lifetimes, however, require the calculation of at least third-order force constants, whose simulation is described in the Section 3.

Figure 8 compares the DFT- and MTP^phonon^-calculated phonon band structures of PE with the sizes of symbols scaled linearly and their coloring logarithmically with the respective phonon lifetimes. Again, the agreement between the two methods is excellent. Interestingly, for both methodologies, one observes an unusual behavior, as phonon lifetimes do not decrease steadily with frequency (as observed usually), but become particularly large for longitudinal acoustic phonons between 12 THz and 15 THz. Following the arguments of Wang et al. [73], who observed this unusual behavior for a single PE chain, it can be explained in the following way: longitudinal acoustic (LA) phonon modes between 5.5 THz and 11 THz have only a low contribution to the thermal conductivity, as the corresponding phonon lifetimes are small. This is the consequence of efficient Umklapp scattering processes involving twisting and transverse modes, e.g., LA → TWA + TA1 (LA = longitudinal acoustic, TWA = twisting acoustic, TA1 = transverse acoustic) [73]. Energy conservation, however, forbids scattering processes of the form LA → TWA + TA1 for phonons with frequencies above ~12 THz, as then, the LA phonons have more than twice the maximum energy of the TA and TWA modes. As a consequence, when only considering three-phonon processes a steep increase in phonon lifetime between 12 THz and 15 THz is obtained. The phonons of a single PE chain deviate in some respects from that of the PE crystal studied here. For example, there are no optical modes below 20 THz [73]. Moreover, for a single PE chain, the twisting mode has zero frequency at the Γ-point, and is, thus, an acoustic mode, whereas, in a PE crystal, the twisting mode has a nonzero frequency at the Γ-point and is, thus, an optical mode. Despite these differences between the single PE chain and the PE crystal, the core argument of Wang et al. concerning energy conservation can also be applied in the case of PE crystals.

With all ingredients in hand, the anisotropic thermal conductivity of PE can be calculated using Equation (1). The results for the full BTE and for the BTE in the RTA are listed in Table 4. This table also compares the results for the “best” MTP^phonon^ with the mean values for all five parametrized versions of MTP^phonon^. A comparison to the results obtained with the “best” MTP^MD^ is prevented by the above-mentioned imaginary frequency phonons calculated with this MTP. The overall agreement between the “best” MTP^phonon^ and the DFT result is again excellent, especially in the chain direction. The observation that *κ_zz_* differs by only 1 Wm^−1^K^−1^ must be considered as a fortunate coincidence. Still, the values for the arithmetic mean of the thermal conductivities for all five parametrized variants of MTP^phonon^ also agree very well with the DFT results. Interestingly, as observed for the lattice constants, the standard deviations of the thermal conductivities calculated with the five MTPs are again similar to the deviations of the MTP means from the DFT results.

To provide more quantitative analyses of the excellent agreement between the DFT and MTP^phonon^ calculated thermal conductivities (beyond the phonon lifetimes indicated in the phonon band structures in Figure 8), Figure 9 compares the contributions of phonons in specific frequency ranges. A complication in this context is that the apparent deviations, to some extent, depend on the way the frequency ranges are chosen: too large frequency ranges allow for an efficient cancellation of positive and negative deviations, while too small frequency ranges suffer from the finite sampling of reciprocal space in conjunction with the possibility that minor shifts of phonon frequencies can change the frequency ranges that specific phonons are associated with. Nevertheless, Figure 9 clearly demonstrates that the overall trends in the contributions of phonons in specific frequency ranges agree rather well between MTP^phonon^ and DFT. A general trend is that, for phonons below ~10 THz, the MTP^phonon^-calculated contributions are somewhat larger, while the opposite is the case for higher-lying phonons. In view of the near perfect agreement of the phonon bands in Figure 7a, we attribute this to minor quantitative deviations in the calculated phonon lifetimes. Phonons above the band gap starting at ~16 THz do not play a role in the thermal conductivity.

The values of the thermal conductivity reported here compare reasonably well to the results described in the literature for applying very similar DFT-based theoretical approaches [73,74]. This is sufficient for the purpose of benchmarking the performance of the MTPs, but the used approach for obtaining *κ* suffers from several conceptual shortcomings. For one, phonon tunneling effects [77] are neglected (both for the DFT and the MTP^phonon^ calculations). These primarily affect small thermal conductivities and, thus, in the present context, only the values for transport perpendicular to the polymer chains. Moreover, thermal expansion effects are not considered, i.e., for calculating thermal conductivities at room temperature, molecular-dynamics-calculated unit cells could be used (see also Section 2.7). Furthermore, effects like frequency renormalization and especially higher-order phonon scattering processes are neglected, which can distinctly reduce the phonon lifetimes above 12 THz due to a substantial increase in the allowed scattering processes. Also, the application of the relaxation time approximation within the BTE affects the results. All these aspects can be relevant for the final values of the thermal conductivity obtained from phonon properties, but they do not directly influence the benchmarking of the MTPs. Another aspect, especially when high-frequency phonons with appreciable lifetimes exist, is that molecular dynamics simulations would suffer from the classical statistics typically assumed for phonon occupations. This affects NEMD and AEMD simulations, as well as the phonon lifetimes obtained in Fourier transforms of equilibrium molecular dynamics runs. A further complication in NEMD and AEMD studies is that the mathematical form of the finite-size extrapolation is not uniquely defined. For systems with thermal conductivities as large as those obtained for PE and PT, this poses a problem, even when extending the simulations to supercells containing more than 100,000 atoms and considering several million time steps. In passing, we note that we performed such simulations with the MTPs discussed here on a contemporary supercomputer. A detailed discussion of all these aspects becomes rather involved and, thus, would seriously distract from the scope of the present paper. Therefore, we will address them in a forthcoming manuscript. In short, there it will be shown that, when considering all aforementioned effects, consistent values of the thermal conductivities for BTE and MD-based simulations are obtained with deviations in the range of a few percent for the best simulations of a specific type.

### 2.7. Studying the Thermal Expansion of PE

To calculate thermal expansion, MD simulations in an *NPT* ensemble at specific temperatures are performed, as described in Section 3 (with convergence tests provided in Appendix A). As such simulations are not feasible over sufficiently long times when employing converged ab initio methods, we will benchmark the performance of MTPs only against temperature-dependent scattering experiments, which, to the best of our knowledge, only exist for PE. Thus, we will (again) focus on this material in the present section. Stability issues for MD at elevated temperatures prevent simulations with MTP^phonon^ at relevant temperatures (see Section 2.2). Therefore, the focus here is on simulations employing MTP^MD^. Our results are compared to experiments by Davis et al. [78], Shen et al. [55], and Takahashi [53] in Figure 10. To better assess the following comparison, the peculiarities of the experimentally determined thermal expansion of PE are discussed first. The thermal expansion of PE is highly anisotropic, due to the dominance of vdW bonds perpendicular to the chain axis (***a*** and ***b*** vectors) and covalent bonds along the chain (***c*** vector). Lattice vector ***a*** (c.f., Figure 1a) significantly expands with temperature by around 0.3 Å from 100 K to 300 K [78], such that its evolution dominates the change in the unit cell volume. The expansion of ***b*** is about an order of magnitude smaller than that of ***a***. In stark contrast to ***a*** and ***b***, PE’s lattice vector ***c*** (along the chain) slightly shrinks with an increasing temperature. This means that, in the experiment by Davis et al. [78], PE has a negative expansion coefficient in that direction with the overall change between 100 K and 300 K being as small as 0.006 Å. In the experiment by Shen et al., the change in the length of ***c*** between 10 K and 297 K could not even be resolved [55]. Each of the five variants of MTP^MD^s that we parametrize captures these trends (see Appendix A). That is, they all show positive expansions of ***a*** and ***b*** that fit the experiments well, and a small negative expansion of ***c***.

In Figure 10, our calculations are reported in two ways: for one, we ran the MD simulations for each of the five MTP^MD^ and took the mean and standard deviation for the lattice constants. These are shown as blue data points with error bars. Additionally, the violet datapoints represent the results for the “best” variant of MTP^MD^. The calculated thermal expansion of lattice vectors ***a*** and ***b*** (vdW-bonding directions) agree well with experiments. For ***c,*** the agreement appears to be worse, with a seemingly larger offset in the absolute value. This is, however, mostly a consequence of the very small magnitudes of the changes in ***c***. This means that, also in the ***c***-direction, the MTP^MD^ correctly predicts that there is virtually no change with temperature. For example, the calculated 100 K lattice constant |***c***| differs by only 0.003 Å (equaling 0.1%) compared to the “32 days” sample from Davis et al. [79]. For the sake of comparison, in Appendix A, a plot of the thermal expansion calculated with the AIREBO potential is contained. Not unexpectedly, that potential fares much worse than the MTPs: it yields lattice constants strongly deviating from experiments and predicts a wrong trend for the evolution of the lattice constant ***c*** with temperature. Finally, it should be noted that the agreement between theory and the experiments for the thermal expansion of PE is better than what we observed for metal–organic frameworks, potentially due to the ionic character and porous nature of these systems [29]. One also must not forget that, even when using ab initio methods, accurately simulating thermal expansion is far from trivial and prone to serious errors, with even minor changes in the used DFT functionals potentially changing the signs of the expansion coefficients [79]. This suggests that an unknown, but potentially significant fraction of the remaining deviation between theory and experiments in Figure 10 is due to shortcomings of the DFT methodology used for the learning of the MTPs, rather than due to the nature of the MTPs and the employed parametrization process.

### 2.8. Energy, Force, and Stress in Molecular Dynamics

As a final step, to test the accuracy of the MTPs in a context intrinsically relevant for MD simulations, we benchmark the performance of the “best” variant of MTP^MD^ in predicting the total energies, forces, and stresses. This is conducted for the DFT-calculated structures generated during the validation active learning run described in Section 2.2. Notably, none of these data have been used in the parametrization process of the MTP. Figure 11 contains bar plots of the differences between the MTP- and DFT-calculated energies, forces, and stresses. The errors of the MTP are generally very low: for total energies significantly below 1%, for forces mostly below 5%, and for stresses also usually below 5%, albeit with a second maximum in the histogram at a deviation around 10% (as shown in Figure 11a,c,e). To put the MTP accuracy into perspective, we compared the deviations between the MTP and DFT results to those between the AIREBO and DFT data in the right panels of Figure 11b,d,f. There, much larger scales for plotting ranges of energy, force, and stress errors are necessary to even display the AIREBO data. These data show that the MTP^MD^ vastly outperforms the AIREBO potential: for the energies, the RMSD of MTP^MD^ is more than two orders of magnitude (factor of 308) smaller than for the AIREBO potential (0.072 meV/atom vs. 22 meV/atom), for the forces, the difference amounts to nearly two orders of magnitude (factor of 64) with RMSD^MD^ values of 0.011 eV/Å vs. 0.72 eV/Å, and for the stress errors, the deviation is similar (factor of 32) with 0.26 kbar vs. 8.3 kbar.

### 2.9. Speed Gain Due to Use of Moment Tensor Potentials

A comprehensive assessment of the speedup by using MTPs instead of employing DFT is futile, as it depends on a variety of aspects, including the nature of the studied system, the number of atoms in the unit cell, the employed computer architecture, and the chosen degree of parallelization—e.g., the number of cores used in the simulations, the details of the DFT approach (including aspects like the chosen basis sets, the *k*-point sampling, convergence criteria, etc.). Also, the codes used for the specific calculations play an important role. Still, to qualitatively assess the overall situation, a specific example shall be discussed here (with details on the used computer architecture and on the MTP, as well as on the DFT simulations provided in Appendix A and in the Section 3): to calculate the phonon band structure of PE, the converged supercell contains 432 atoms. Starting from an already parametrized MTP, the MTP^phonon^ calculation to obtain the bands (shown in Figure 7) was roughly 10^5^ times faster than the DFT simulation (see Appendix A on how we arrived at this estimate). While the computational complexity of DFT calculations scales cubically with the number of electrons in a naive DFT implementation, MTP simulation times scale linearly with the number of atoms. Therefore, MTPs have an even larger advantage for larger cell sizes, e.g., for converging NEMD and AEMD simulations to obtain thermal conductivities. The exact number of atoms that need to be considered in such calculations strongly depends on the magnitude of the thermal conductivity, but it typically reaches several ten thousand atoms; for materials with high thermal conductivities like PE, in the chain direction, this is potentially even more. For such a case, one can estimate a speedup on the order of nearly 10^10^ (needless to say that the exact value cannot be determined, as converged DFT simulations are entirely infeasible for such a number of atoms).

What still needs to be accounted for is the computational effort for parametrizing the MTP, which has to be performed separately for every considered material. For generating the training structures for MTP^phonon^, we used the same supercell as that for the phonon calculations. For PE, this resulted in 129 DFT-based force calculations in the active learning run between 15 K and 100 K. Training one MTP^phonon^ using these reference data requires a similar amount of resources due to the very high level of the MTP^phonon^ (see Appendix A). Thus, in total, the parametrization of a single variant of MTP^phonon^ costs the equivalent of ~250 DFT calculations (here, with 432 atoms). For parametrizing a single MTP^MD^, 550 ab initio MD steps were performed, but since the effort of parametrizing the lower-level MTP^MD^ is much lower, the overall effort is ~2.5 times as large as that for the MTP^phonon^. This means that, for obtaining quantities that are relatively easy to calculate, parametrizing an MTP is not necessarily advisable. For example, due to the high symmetry of the material and the small primitive unit cell, for calculating the phonon band structure of PE, only 12 DFT calculations are needed. This means that, for PE, calculating the phonon band structure with an MTP^phonon^ is approximately twenty times as expensive as calculating the phonon band structure with DFT in the first place due to the MTP parametrization. For more complex materials requiring a much larger number of DFT calculations for obtaining phonon band structures (but a similar number of DFT reference calculations for the MTP parametrization), this gap diminishes or even closes. This, we have, for example, observed for a variety of metal–organic frameworks.

MTPs become much more efficient than DFT when phonon lifetimes need to be calculated, because, in that case, even for PE, 8000 DFT calculations are necessary. This becomes more of an issue for larger systems, where the number of necessary DFT calculations becomes insurmountable (around 50,000 and 360,000 for PT and P3HT, respectively). Even more significantly, performing NEMD calculations over 5 ns with 0.5 fs time steps requires calculating forces ten million times. This typically involves a much larger unit cell than what is needed for converging phonon properties. These numbers illustrate the main benefit of the presented approach of parametrizing MTPs for polymer materials, namely, that it opens up the possibility for performing new types of calculations, for which DFT would be far too expensive and off-the-shelf, transferrable force fields would be far too inaccurate.

## 3. Methods

### 3.1. Details of the Applied Computational Approach

DFT calculations were performed with VASP version 6.3.0 [44]. Its primary task was to generate training data for parametrizing the MTPs. VASP was employed to calculate elastic constants and (in conjunction with phonopy and phono3py [9,80,81]) to calculate phonon band structures and the thermal conductivity of PE (employing the BTE, as described in the Section 2.6 and Section 3.2). The PBE functional [43], complemented by Grimme’s D3 correction [18,19] with Becke–Johnson damping [20], was employed, as this combination is well established for accurately describing the phonons of organic semiconductors [17,21,82,83]. Following convergence tests for the phonon band-structures shown in Appendix A, the energy cutoff determining the size of the plane-wave basis set was set to 900 eV (PE) and to 700 eV (PT and P3HT). When relaxing cells, the energy cutoff was increased to 1350 eV to mitigate Pulay stress. Geometry relaxations (with a convergence criterion for maximum forces of 0.001 eV/Å) were performed in two steps: first, all degrees of freedom, including the unit cell parameters, were relaxed with the 1350 eV cutoff. Subsequently, the unit cell parameters were kept fixed, and only the atomic positions were relaxed with an energy cutoff of 900 eV (PE) or 700 eV (PT and P3HT). The second step was necessary to relax the atomic positions with the same energy cutoff that was later used in the phonon calculations.

*K*-meshes for describing the electronic wavefunctions were set such that the phonon band structures were converged. We found that phonon calculations with supercells (see below) converged with 1 × 1 × 1 (PE and P3HT) and 1 × 1 × 2 (PT) ***k***-meshes. This defined the *k*-meshes used for the primitive unit cells as 2 × 3 × 6 (PE), 2 × 3 × 4 (PT), and 2 × 1 × 2 (P3HT). The convergence tests of the *k*-mesh can be found in Appendix A. The break condition for the electronic self-consistency loop was set to 10^−8^ eV. Partial occupancies of orbitals were treated with a Gaussian smearing with a width of 0.05 eV.

In the VASP active learning runs, mostly the same settings as those described above were used. Also, in active learning runs, unit cells are distorted, again causing Pulay stress. To mitigate this, an increase in the plane-wave cutoff of at least 30% compared to fixed volume calculations is recommended on the VASP wiki [51]. As large volume fluctuations indeed occurred for P3HT, we increased the energy cutoff in the active learning from 700 eV to 900 eV. As no large volume fluctuations occurred for PE and PT, for these systems, the cutoffs were kept at 900 eV and 700 eV, respectively. As an additional measure to mitigate Pulay stress, the active learning runs were split into smaller sections to reinitialize the plane wave basis sets. For P3HT, the runs were performed from 15 K to 300 K, 300 K to 400 K, and 400 K to 500 K. For PE, the learning runs were split into sections from 15 K to 30 K, 30 K to 150 K, 150 K to 300 K, 300 K to 400 K, and 400 K to 500 K (see discussion in Appendix A). For PT, the run was not split into multiple segments, as the change in the unit cell size was particularly small for that polymer. Supercells for the training data were chosen such that they extended approximately 15 Å long in each direction. This choice generally also yields converged phonon band structures, as detailed below and shown in Appendix A. Correspondingly, a 2 × 3 × 6 supercell was chosen for PE, a 2 × 3 × 2 supercell for PT, and a 2 × 1 × 2 supercell for P3HT. For the active learning MD runs, a Langevin thermostat with a friction of 10 ps^−1^ for all atoms and the unit cell was used. VASP active learning runs are generally performed in an *NPT* ensemble (except for Appendix A, where we describe the influence of using an *NVT* ensemble). For the active learning runs, the fictitious mass of the lattice degrees-of-freedom was set to 1000 amu. The time step was set to 0.5 fs. For training the MTPs geared towards phonon calculations (MTP^phonon^), we performed an *NPT* run from 15 to 100 K for 12,750 time steps (i.e., 150 steps/K). This yielded 129 DFT-calculated training structures for PE, 173 training structures for PT, and 139 structures for P3HT. For the MTP^MD^ training data, the run was continued to 500 K, yielding a total of 72,750 steps (i.e., again with 150 steps/K). This resulted in 550, 490, and 549 DFT-calculated training structures for PE, PT, and P3HT, respectively.

To assess the accuracy of the MTPs, we created a validation set that consisted of independent DFT reference calculations from an MD run at a fixed temperature of 300 K starting from scratch (i.e., disregarding any data from the above-mentioned training). This run lasted for 10,000 time steps in an *NPT* ensemble for PE and P3HT. For PT, because the structure became unstable during the run (i.e., atoms detached from the polymer chains), we had to enforce more DFT steps to make it stable. This was achieved by setting the tag ML_CX to −0.15 (see VASP manual [84]). This tag controlled the threshold that determined whether a DFT calculation was performed and helped to avoid too few training data for runs at constant temperature [51]. As a result of this procedure, we ended up with more DFT reference structures for PT as compared to the other two materials. Thus, the validation sets consisted of 136 structures for PE, 296 for PT, and 182 for P3HT. Some initial configurations are associated with a lower temperature, because it takes some time for the thermostat to reach the desired 300 K. Considering the example of PE, it took around 250 time steps to reach 300 K. We found that this was irrelevant for the calculated RMSD^MD^, whether we cut out these poorly equilibrated structures or not. Therefore, we kept them in the validation set.

The AIREBO potential [71] was used as implemented in LAMMPS, with its functional form given in Appendix A. The scaling factor for the Lennard–Jones term, which defines the cutoff, was set to 3.

### 3.2. Modeling Physical Observables

Harmonic phonon properties were obtained using the phonopy package [9,85]. It uses a finite differences scheme with a displacement distance of 0.01 Å. This requires the calculation of forces on all atoms upon displacing individual atoms. The number of displacements that eventually needs to be obtained can be significantly reduced by considering symmetries. For example, in the case of PE (with 12 atoms in the primitive unit cell), only 12 calculations are necessary. Notably, in such calculations, sufficiently large supercells must be used with details on the supercell convergence contained in Appendix A. For our materials, we typically found well-converged phonon band structures for supercells, with an extent of about 15 Å in each direction. An exception was PT, for which a supercell of 2 × 3 × 2 would be around 15 Å large in each direction. However, for this supercell size, the phonon band structure had a small dip in imaginary values along the Γ-X path close to Γ. To mitigate this dip, a 2 × 3 × 4 supercell was used. Correspondingly, 2 × 3 × 6, 2 × 3 × 4, and 2 × 1 × 2 supercells were used for the phonon calculations of PE, PT, and P3HT. The same supercells were used in the DFT and MTP simulations. As described above, to quantitatively measure the agreement between MTP and DFT, we defined the root mean square deviation (RMSD) between the phonon frequencies of the MTP and DFT, RMSD^phonon^. We calculated the RMSD^phonon^ over the whole Brillouin zone, which we sampled with dense *q*-point meshes of 30 × 44 × 83, 30 × 42 × 30, and 10 × 5 × 9 *q*-points for PE, PT, and P3HT, respectively.

For calculating the anharmonic phonon properties, we employed the phono3py code [9,80], which considers third-order force constants and, consequently, accounts for three-phonon scattering processes. Phono3py was used to calculate the phonon lifetimes and the thermal conductivity via the Boltzmann transport equation (BTE), either directly (keyword “lbte”) or within the relaxation time approximation. Displacements of 0.03 Å were used for calculating these third-order force constants. This is the default in phono3py and was checked for convergence (see Appendix A). The supercell for the third-order force constant calculation, which was performed only for PE, was a 2 × 2 × 3 cell for both the DFT and the MTP calculations. Convergence tests for choosing that supercell size are provided in Appendix A. Brillouin zone integration in phono3py is performed with the default tetrahedron method [86], which has the advantage of being a parameter-free approach. Brillouin zones are sampled with a 10 × 15 × 160 mesh for the RTA calculation. When employing the full BTE, a 10 × 15 × 60 mesh is used for *κ_xx_* and *κ_yy_*, while a 4 × 6 × 160 mesh is used for *κ_zz_*. These different mesh sizes for the respective elements of *κ* were used to reduce computational and memory costs, as a 10 × 15 × 160 mesh for the full BTE calculation would be infeasible due to the excessive demand for memory. Convergence tests for these meshes are provided in Appendix A. Phonopy and phono3py require forces, agnostic to whether they are calculated with VASP or MLIP, therefore, the same phonopy and phono3py settings can be used for both force calculators.

To obtain the elastic stiffness tensors, a finite differences approach was employed [87]. For the DFT calculations, we used an in-house developed code, in which the cell was strained by 1%. This strain distance is converged, as elastic constants do not change significantly whether strains of 1% or 0.1% are used, and change only slightly for strains of 0.01%, as is shown in Appendix A. The contribution from relaxing the ions in the strained cells is approximated from second-order force constants, and the elastic tensor is obtained from the strain–stress relationship [62,87]. This approximation is performed to save computational resources, and we refer to this approach as the “clamped ion method” [88]. This method is identical to the VASP-internal routine (setting IBRION = 6 and ISIF = 3), with the important difference that, in the VASP-internal approach, the POTIM parameter controls both the atomic displacement distance to calculate second-order force constants as well as the strain distance, whereas, in our in-house developed code, these two settings are disentangled. This allowed us to keep the atomic displacement at 0.01 Å for the second-order force constants (calculated via phonopy), while converging the strain distance independently. In fact, we used the force constants for which we already obtained converged phonon band structures. The energy cutoffs were also kept identical to those used in the phonon band structure calculation. A calculation of P3HT with an energy cutoff of 900 eV is given in Appendix A. It shows only small deviations from the calculation with an energy cutoff of 700 eV, thereby suggesting that the energy cutoff is converged for calculating the elastic tensor. For the elastic constants calculations with MTPs in the main paper, we employed the same method (“clamped ion method”) as that with DFT for consistency reasons. However, since the MTPs are orders of magnitude faster than DFT, it is also substantially cheaper to calculate elastic constants with a less approximative method. In this method, after straining the cell by 0.1%, the ionic positions were relaxed and the stress tensor for the strain–stress relationship was directly calculated with the MTPs. We refer to this less approximative method as the “relaxed ion method”, and the results are shown in Appendix A.

Thermal expansion was calculated in conventional MD simulations employing the MTPs. MD was performed in LAMMPS [49] (Version 2nd July 2021) in combination with the LAMMPS–MLIP interface (Version 2) [26]. A Nose–Hoover thermostat with a damping parameter of 0.1 ps and a Nose–Hoover barostat with damping parameter of 1 ps were used. A time step of 0.5 fs was chosen, which is consistent with previous works, where time steps between 0.25 fs and 2 fs were employed for (non-equilibrium) molecular dynamics simulations of PE [14,89,90]. For calculating thermal expansion, an orthorhombic 4 × 6 × 12 supercell was used, which has an extent of approximately 30 Å in all dimensions. An *NPT* run was performed for 800 ps at each set temperature, during which the lattice parameters were tracked (see Appendix A for a convergence test). Ignoring the first 10 ps to account for equilibration effects, the lattice parameters were then averaged over the remaining time of the run.

## 4. Conclusions

Machine-learned potentials are a powerful tool in materials’ modeling, as they allow simulating structural and dynamic properties with close to DFT accuracy at computational costs that are reduced by many orders of magnitude. This is illustrated here for moment tensor potentials (MTP), which are parametrized for specific materials, with a focus on phonon-related properties and molecular dynamics (MD) simulations. In particular, we discuss how to optimally parametrize MTPs and how to benchmark their performance.

For the parametrization, it turned out to be useful to distinguish between two distinct use cases: as the first use case, MTPs were applied for finite difference calculations, where all atoms remained close to their equilibrium positions. These included calculations of phonon band structures and the thermal conductivity via the Boltzmann transport equation. The second use case was molecular dynamics calculations at elevated temperatures (e.g., room temperature), which can be employed to, for example, simulate thermal expansion. In such simulations, significantly larger atomic displacements occur. This calls for establishing separate protocols for these two scenarios. In that regard, a crucial aspect is the suitable selection of training data, which, in our case, were generated during active learning molecular dynamics simulations. The magnitudes of atomic displacements covered by the training data were determined by the considered temperature ranges in the active learning runs. In this context, we found it to be advantageous to parametrize the MTPs for the phonon calculations using reference structures generated at low temperatures (e.g., between 15 K and 100 K), while a force field for MD calculations had to be parametrized against reference data, whose generation involved higher-temperature reference structures (generated, e.g., between 15 K and 500 K). In particular, we found that MTPs trained on low-temperature data were twice as accurate in phonon calculations as their counterparts trained also on structures generated at higher temperatures. The opposite was observed for MD simulations, where MTPs containing only low-temperature data in their parametrization even turned out to be unstable.

Further strategies for improving the accuracy are increasing the level of the MTP and splitting chemically inequivalent atoms into multiple atom types. Both of these strategies increase the number of parameters of the MTPs, which, in the case of the level, comes at the cost of considerably increased computational costs. For the phonon calculation, computation time is not such a serious concern, as the size of the considered supercells and the number of force calculations are not excessively large. Therefore, a higher level can be used than for MTPs that are meant for molecular dynamics calculations. The situation is different when increasing the number of atom types. This only increases the time needed for the parametrization, which leads us to suggest to, by default, treat each atom in a chemically different environment as a separate species. These considerations yield two optimum styles of MTPs for the two described use cases: MTP^phonon^, which is a level 26 MTP trained against active learning data generated between 10 K and 100 K, and MTP^MD^, as a level 22 MTP trained on data generated between 10 K and 500 K. The two types of MTPs were benchmarked in terms of their ability to describe unit cells, phonon band structures, and elastic constants for polyethylene (PE), polythiophene (PT), and poly-3-hexylthiophene (P3HT). For PE, also the thermal conductivity (via the Boltzmann transport equation), the thermal expansion (via MD), and forces, energies, and strains during an MD run were assessed. All these tests (i) testify to the exceptional accuracy of system-specific MTPs (relative to DFT data and experiments) and (ii) show that the MTP variants parametrized for the specific use cases display a distinctly improved performance.

All this was achieved with the MTP simulations being between five and (an estimated) ten orders of magnitude faster than the DFT calculations using the methodology employed for their parametrization. For calculating relatively simple quantities (like structural parameters and phonon band structures) of relatively simple materials (like PE), this increase in speed is, however, more than offset by the computational efforts needed to parametrize the MTPs. This is clearly not the case when calculating computationally more demanding quantities, like phonon lifetimes or MD trajectories (for obtaining thermal expansion coefficients or thermal conductivities).

Many of these calculations only become possible by machine-learned potentials, as ab initio methods like DFT would be far too expensive, and off-the-shelf force fields would be too inaccurate. Thus, we are confident that the current manuscript will have a profound impact on the future simulation of the structural and dynamic properties of polymers and will, for example, lay the basis for the quantitatively reliable simulation of their heat transport properties employing molecular dynamics and/or lattice dynamics simulations.

## Figures and Tables

**Figure 2 molecules-29-03724-f002:**
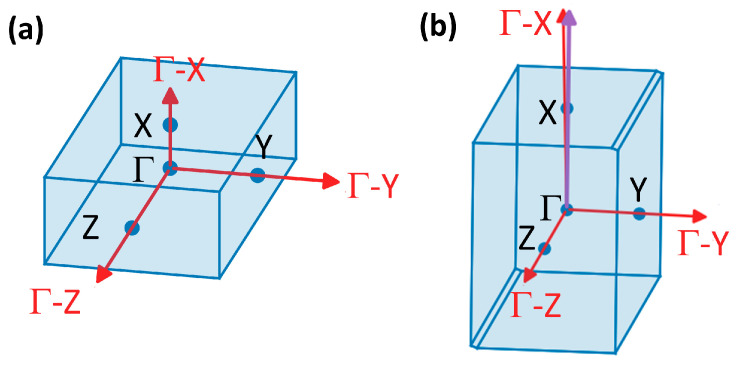
First Brillouin zones of PE (**a**) and P3HT (**b**). The high symmetry points X, Y, Z, and Γ are labelled. The paths Γ-X, Γ-Y, and Γ-Z are shown with red arrows and correspond to the phonon band structure paths shown in Section 2.5. In all systems, Γ-Z is parallel to the polymer axis. The Bravais lattice is primitive orthorhombic for PE (and also for PT—not shown here) and primitive monoclinic for P3HT. Thus, in P3HT, the Γ-X direction (red arrow) is slightly inclined relative to the real space direction *x* (purple arrow).

**Figure 3 molecules-29-03724-f003:**
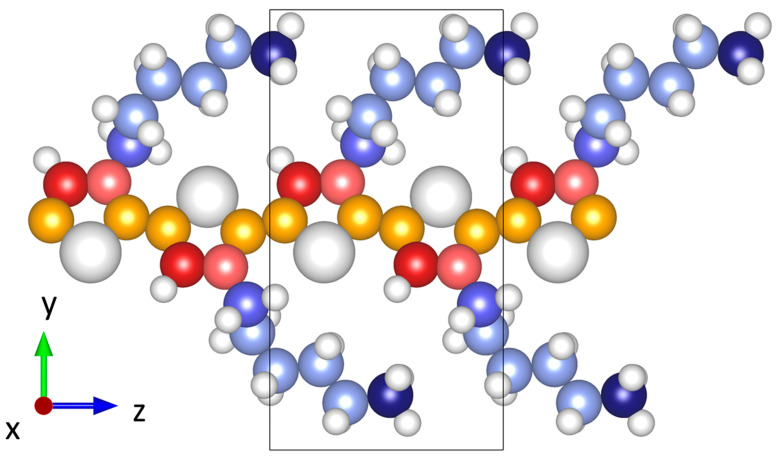
Illustration of how the carbon atoms of P3HT can be split into at most six atom types. Sulfur and hydrogen atoms are white. Carbon atoms with up to six different chemical environments can be distinguished: sp^2^-hybridized carbon atoms bonded to two other carbon atoms and one sulfur (orange), bonded to two carbon atoms and one hydrogen (dark red), and bonded to three carbon atoms (light red); furthermore, sp^3^-hybridized carbon atoms bonded to the sp^2^-hybridized backbone (medium bright blue), sp^3^-hybridized carbons in the middle of the sidechain (light blue), and at the end of the side chain (dark blue). The unit cell is shown as a black box and contains two P3HT chains, but for the sake of clarity, only one is shown here.

**Figure 4 molecules-29-03724-f004:**
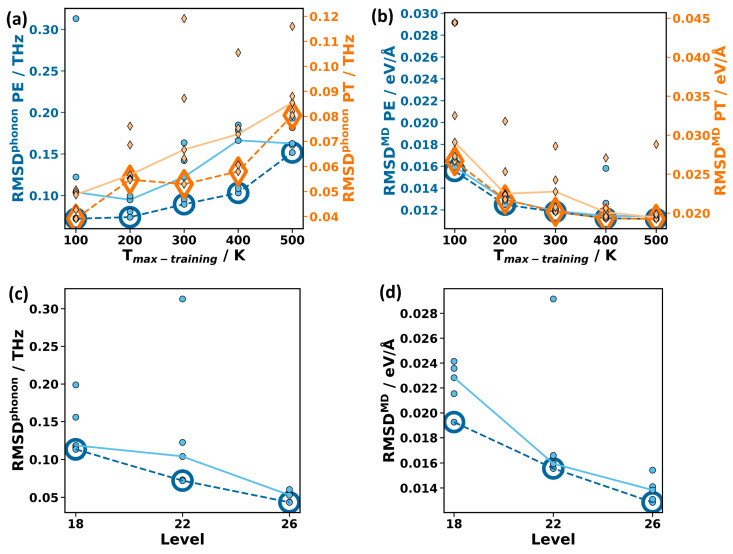
Influence of the training data and the chosen level on the accuracy of MTPs. Accuracy is measured by the root mean square deviation (RMSD) between MTP- and DFT-calculated phonon frequencies sampled in the whole Brillouin zone up to 12.5 THz (RMSD^phonon^) and by the RMSD of forces on atoms calculated for validation structures generated in a 300 K active learning MD run (RMSD^MD^). Panels (**a**,**b**) show RMSD^phonon^ and RMSD^MD^ for different training data sets for PE (blue) and PT (orange) for the MTP level set to 22. The values on the *x*-axis denote the maximum temperature in the generation of the respective reference data during active learning runs (for details see main text). Panel (**c**) shows RMSD^phonon^ for PE for different MTP levels (generated with 15–100 K training data), and panel (**d**) RMSD^MD^ for PE for different MTP levels (again generated with 15–100 K training data). For each data set/level, five MTPs are parametrized and the associated RMSD values are shown as small, filled symbols. Datapoints for MTPs with median RMSD^phonon^/RMSD^MD^ are connected by solid lines, while data for the “best” MTPs are denoted by large, dark, open symbols and are connected by dashed lines. The numerical values for these plots are provided in Appendix A of the Appendix A.

**Figure 5 molecules-29-03724-f005:**
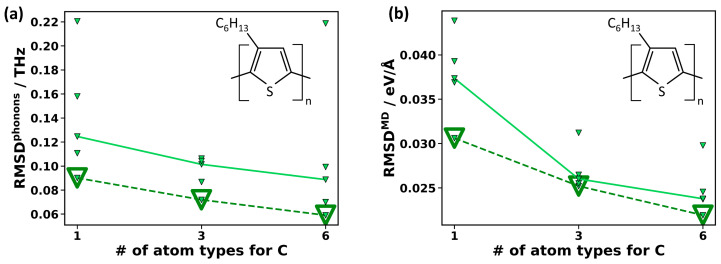
Influence of increasing the number of atom types on MTP accuracy for P3HT. Panel (**a**) shows RMSD^phonon^, panel (**b**) RMSD^MD^ for treating all carbon atoms in P3HT the same (one atom type), when distinguishing between sp^3^-hybridized carbons and sp^2^-hybridized ones, bonded either only to other carbons or to one sulfur and two carbons (three atom types), and when distinguishing between all chemically inequivalent carbons (six atom types), as shown in Figure 3. The level of the MTP is set to 22 and the training data are taken from an active learning run between 15 K and 500 K. For each number of atom types, five MTPs are parametrized and the associated MTPs are shown as small, filled symbols. MTPs with median RMSD^phonon^/RMSD^MD^ are connected by solid lines, while data for the “best” MTPs are denoted by large, dark, open symbols and are connected by dashed lines. Numerical values associated with these plots are given in Appendix A of the Appendix A.

**Figure 6 molecules-29-03724-f006:**
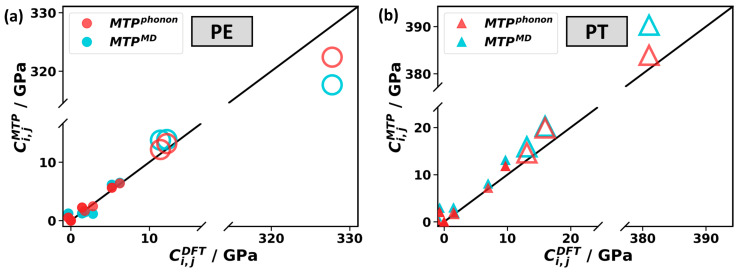
Independent elements of the elastic tensor calculated with MTP^MD^ and MTP^phonon^ compared to the results obtained with DFT. All simulations are performed applying the “clamped ion method” as explained in the main text. The investigated materials are (**a**) PE and (**b**) PT. The black line with a slope of 1 indicates perfect agreement between MTP and DFT. The displayed data are obtained with the “best” MTPs as defined at the end of Section 2.2. The large open symbols represent *C_xx_*, *C_yy_*, and *C_zz_*, while the small symbols represent the other independent components of the elastic tensor. An analogous plot containing the mean values over elastic constants calculated with all parametrized (typically five) MTPs, and corresponding error bars can be found in Appendix A.

**Figure 7 molecules-29-03724-f007:**
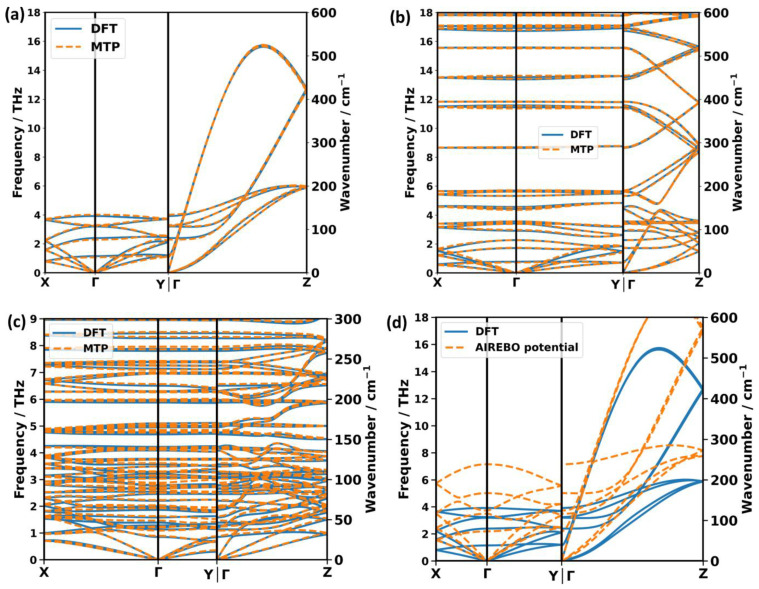
Panel (**a**–**c**) phonon band structures calculated with DFT (solid blue lines) and the “best” MTP^phonon^ (dashed orange lines) in the low-frequency region. Materials are (**a**) PE, (**b**) PT, and (**c**) P3HT. The frequency range for P3HT is half that for PE and PT, such that the many bands are resolved more clearly. A corresponding plot for the extended frequency range is given in Appendix A. Panel (**d**) compares the phonon band structures of PE calculated with DFT and with the AIREBO potential.

**Figure 8 molecules-29-03724-f008:**
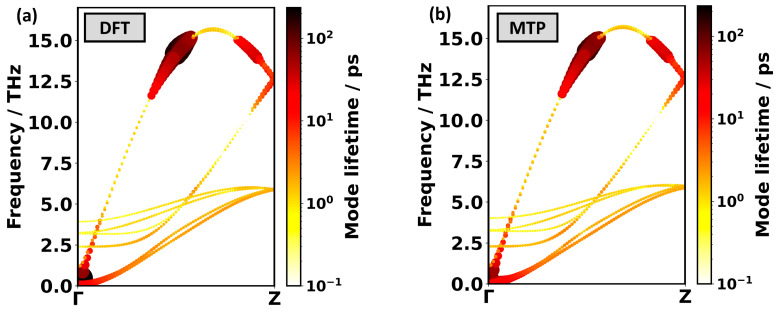
Phonon bands along the chain direction in PE for which the lifetimes of the phonons are encoded by the size of the symbols and by their colors. They have been calculated with (**a**) DFT and with (**b**) the “best” MTP^phonon^. While the color scale is logarithmic, the areas of the spherical symbols are scaled linearly with the phonon lifetimes. Data points depicted as smaller circles are plotted in front, such that they are not hidden by larger circles. We note that a similarly good agreement is obtained for the other four variants of MTP^phonon^ that were trained.

**Figure 9 molecules-29-03724-f009:**
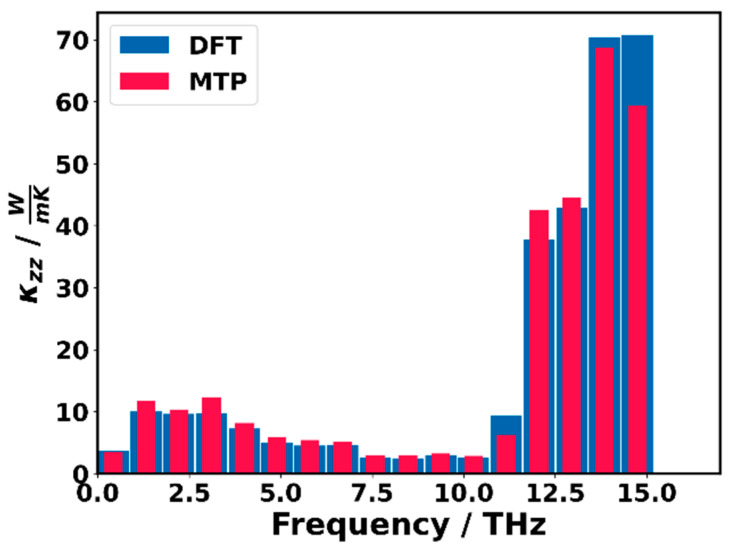
Contributions to the thermal conductivity of PE along the chain (*κ_zz_*) in the RTA for different frequency ranges. Data calculated with DFT are compared to those obtained with the “best” MTP^phonon^.

**Figure 10 molecules-29-03724-f010:**
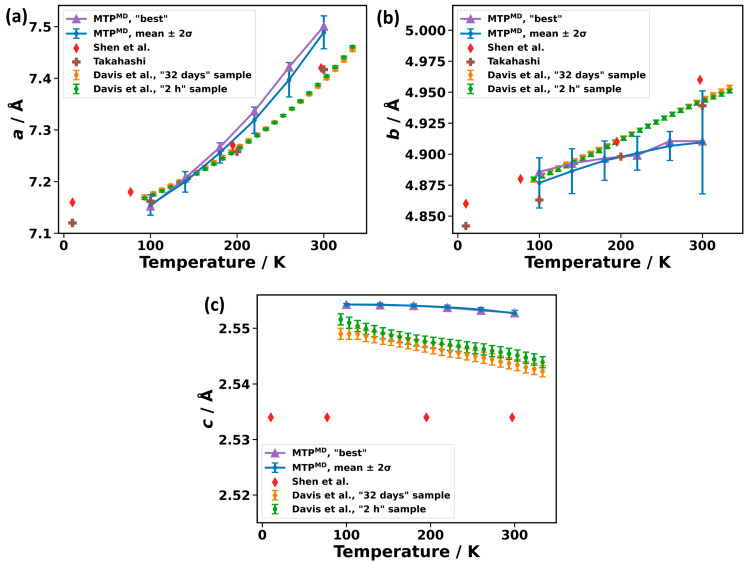
Thermal expansion of PE along its lattice vectors (**a**) ***a***, (**b**) ***b***, and (**c**) ***c***. The purple line labelled ’MTP^MD^, “best”’ is the MTP^MD^ with lowest RMSD^MD^. The blue line labelled “MTP^MD^, mean ± 2σ” is the mean from five molecular dynamics runs each with a different MTP^MD^; whereby the error bars are two times the standard deviation (95% confidence interval). Experimental data from Davis et al. [78], Shen et al. [55], and Takahashi [53] are shown. Davis measured two samples, which they called “32 days” and “2 h” [78]. These names refer to how long the samples remained in constant temperature baths during their production. Shen et al. measured PE single crystals.

**Figure 11 molecules-29-03724-f011:**
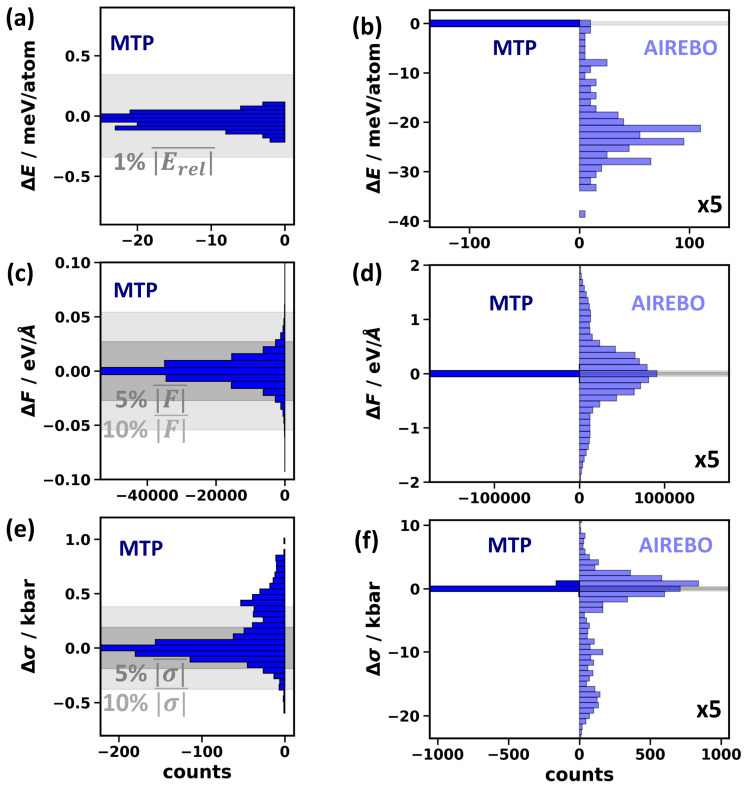
Histograms of the deviations to DFT reference data for equilibrium energies, ΔE, forces on individual atoms, ΔF, and stresses, Δσ. The compared quantities were obtained with the “best” MTP^MD^ (dark blue shading) and with the AIREBO potential (light blue shading). Panels (**a**,**c**,**e**) show the data for the “best” MTP^MD^. Panels (**b**,**d**,**f**) show the same data for MTP^MD^ (albeit on a different scale) and compare them to data obtained with the AIREBEO potential. The number of counts was scaled by a factor of five for the AIREBO potential to improve visibility. Gray-shaded areas represent ranges where errors are within 5% and 10% of the absolute DFT values for forces and stresses, and within 1% for energy differences.

**Table 1 molecules-29-03724-t001:** Comparison of the lengths of the lattice vector ***a***, ***b*,** and ***c*** and the monoclinic angle (between the ***a***- and ***b***-vectors), *α*, calculated with MTPs and with DFT by minimizing the energy, as detailed in Section 3. For the two types of MTPs (MTP^phonon^ and MTP^MD^), the results for the “best” MTP (with the lowest RMSD^phonon^/RMSD^MD^ values) are given. Additionally, the arithmetic mean values for the five parametrized potentials for each system are reported. The uncertainties of the mean MTP values only refer to the statistical errors caused by the MTP parametrization, as they are the standard deviation for the data from the five MTPs parametrized with different seeds. For MTP^phonon^ in the case of P3HT, no uncertainties are reported, as there only two MTPs are parametrized due to the associated high computational costs associated with generating level 26 potentials for such a complex material. Experimental data are taken from the literature. Temperatures at which the experiments are performed are given in parentheses (with RT meaning room temperature) [41,52,53,54,55]. For the specific conformation of P3HT studied here (see Section “Structures of Polyethylene, Polythiophene and Poly(3-Hexylthiophene-2,5-Diyl) (P3HT)”), no experimental data are found.

	*a* [Å]	*b* [Å]	*c* [Å]	*α* [°]
Polyethylene				
DFT	7.074	4.853	2.554	90
MTP^phonon^, “best”	7.062	4.847	2.5543	90
MTP^phonon^, mean	7.060 ± 0.003	4.849 ± 0.002	2.5543 ± 0.0001	90
MTP^MD^, “best”	7.050	4.853	2.5541	90
MTP^MD^, mean	7.052 ± 0.013	4.849 ± 0.015	2.5542 ± 0.0002	90
Experiment (4 K) [52]	7.121	4.851	2.548	90
Experiment (10 K) [53]	7.120	4.842	-	90
Experiment (10 K) [55]	7.16	4.86	2.534	90
Polythiophene				
DFT	7.530	5.542	7.785	90
MTP^phonon^, “best”	7.467	5.508	7.782	90
MTP^phonon^, mean	7.472 ± 0.003	5.506 ± 0.002	7.782 ± 0.001	90
MTP^MD^, “best”	7.500	5.488	7.782	90
MTP^MD^, mean	7.449 ± 0.043	5.524 ± 0.027	7.781 ± 0.001	90
Experiment ^1^ [54]	7.80	5.55	8.03	90
Experiment (RT) [41]	7.79	5.53	-	-
P3HT				
DFT	7.575	14.731	7.816	88.75
MTP^phonon^, “best”	7.561	14.705	7.815	88.81
MTP^phonon^, mean	7.564	14.703	7.816	88.74
MTP^MD^, “best”	7.573	14.657	7.816	88.7
MTP^MD^, mean	7.569 ± 0.007	14.675 ± 0.021	7.816 ± 0.001	89.13 ±0.46

^1^ Temperature not specified.

**Table 3 molecules-29-03724-t003:** RMSD^phonon^ values for the “best” and median MTPs obtained when comparing DFT- and MTP-calculated phonon band structures for a homogeneous sampling of reciprocal space. The values are given in THz and in parentheses also in cm^−1^. Note that the values reported here are obtained by fixing the unit cell to the DFT-optimized parameters to avoid simultaneously assessing the ability of the MTPs reproducing unit cell parameters and phonon properties. RMSD^phonon^ values for MTP optimized unit cells are contained in Appendix A. Due to the high computational cost, only two MTP^phonon^ are parametrized for P3HT. Therefore, there is no median MTP in that case.

	PE [THz and (cm^−1^)]	PT [THz and (cm^−1^)]	P3HT [THz and (cm^−1^)]
MTP^phonon^, “best”	0.043 (1.43)	0.029 (0.97)	0.036 (1.20)
MTP^phonon^, median	0.053 (1.77)	0.032 (1.07)	-
MTP^MD^, “best”	0.152 (5.07)	0.080 (2.67)	0.059 (1.97)
MTP^MD^, median	0.163 (5.44)	0.086 (2.87)	0.089 (2.97)

**Table 4 molecules-29-03724-t004:** Thermal conductivity of PE at 300 K, calculated with the BTE employing DFT and MTPs. RTA refers to the use of the relaxation time approximation. “Full BTE” refers to BTE calculations disregarding the RTA. For the result of the mean MTP, the arithmetic mean is taken over the thermal conductivities calculated with all five MTPs. The uncertainty is the standard deviation calculated for the thermal conductivities obtained with these five MTPs.

	*κ_xx_* [Wm^−1^K^−1^]	*κ_yy_* [Wm^−1^K^−1^]	*κ_zz_* [Wm^−1^K^−1^]
RTA			
DFT	0.54	0.46	306
MTP^phonon^, “best”	0.71	0.54	307
MTP^phonon^, mean	0.67 ± 0.09	0.53 ± 0.08	292 ± 33
Full BTE			
DFT	0.52	0.47	398
MTP^phonon^, “best”	0.71	0.58	408
MTP^phonon^, mean	0.66 ± 0.09	0.58 ± 0.09	388 ± 40

## Data Availability

All relevant data can be downloaded from the TU Graz data repository: https://doi.org/10.3217/7s1ce-ss195.

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
