# Peer review of "Designing Accurate Moment Tensor Potentials for Phonon-Related Properties of Crystalline Polymers"

_molecules, 2024, doi:10.3390/molecules29163724_

Round 1

Reviewer 1 Report

Comments and Suggestions for Authors

This is a very comprehensive work on the application of machine-learned potentials (MLPs) to the study of phonon related properties of crystalline polymers. Different ways of training dataset generation and atom-type encoding are compared, which are useful for reducing DFT calculations and enhancing the training accuracy, although at the sacrifice of reducing the generality of the models. The results are carefully discussed and the conclusions are well supported. I would like to see the publication of the paper soon, after the following question is addressed.

The authors demonstrated a high force accuracy in the test dataset, but why stop here? I would like to see a comparison of the thermal conductivity values from NEMD (or AEMD) and BTE, which is required to more properly demonstrate the claimed high accuracy and efficiency of the MLPs in realistic (large size and long time) MD simulations.

Reviewer 2 Report

Comments and Suggestions for Authors

Authors present a modeling study on the design of MTPs to calculate selected properties of polymers in the crystal phase. The work is very interesting and could attract the interest of both researchers active in simulation and experimentalists. The methodology is exhaustively presented (except some aspects of the molecular model, see below), including information accessible as S.I. Subject to addressing the comments and questions raised above the work should be publishable in Molecules.

) I have some trouble understanding which molecular model is used to represent the three different polymers even after accessing the very extensive information available in S.I. Focusing on the simplest of the three, PE, how, chain connectivity is implemented? How non-bonded interactions are treated? Finally, what is the degree of polymerization, in other words, the number of carbon atoms, per chain? The molecular weight of the simulated polymers should affect profoundly the observed physical quantities if the polymeric plateau is not reached. Also, one wonders how accurately basic quantities like density, or the characteristic ratio of PE are captured even in the melt state (temperatures as high as 500 K could not correspond to crystalline PE).

To make the manuscript more accessible I would suggest to move the methods section as early as possible including a more detailed description of the molecular model adopted.

) Given the discussion on carbon atoms being treated differently according to the bond chemisty in the PE chain are the carbon atoms of CH2 different compared to the ones at the ends (CH3) ?   

) I have some concerns about the length of the manuscript. Perhaps authors could consider placing additional material in S.I.

) One technical question is how the radius of 5A is chosen for the local environments. Is this range the same for all three polymers studied here? Is there a general rule on how to select it? Is it based on the type of interactions? Could one consider different approaches like for example through a Voronoi tessellation?

Furthermore, does this radius depend on the lattice spacing of the corresponding formed crystal?

) Again, on the technical implementation of MTP lower an advice is given (lines 229-230) to use lower-level approach to enhance the speed of MTP for MD simulations. However, it is not apparent how general is this statement. Additionally, authors use a radial basis size which ranges from 10 to 26. However, how this increase affects the computational performance of the approach is not demonstrated. If this is reported in the past a short comment should be enough. Otherwise, the corresponding CPU times could be presented. Finally, when an extended radial basis should be used? What would be the practical ceiling in that case?

) As a philosophical question: Would the present methodology be applicable by feeding information required for MTP not from MD simulation data but from MC ones? Polymer systems, even melts, are notoriously difficult in their equilibration once the entangled regime is reached so that MD is too slow to efficiently sample the system especially with respect to long-range characteristics. A potential can thus predict accurately some aspects of the short-range and local behavior but could in principle fail to capture correctly, quantities like polymer stiffness or global size. I wonder if for example structural and conformational results as a function of temperature (like the ones presented in J. Phys. Chem. B 113 442 (2009)) could be of any use to such an approach (clearly there is no corresponding dynamical information available but could be extracted from analogous MD simulations).

) Abstract: “Resolving this dilemma”, it is not exactly a dilemma as none of the solutions is adequate.

Round 2

Reviewer 1 Report

Comments and Suggestions for Authors

I undersatand that comparing BTE and MD methods in detail are involved, but I don't undestand why NOT calculating thermal conductivity using MD method(s) at all, if the authors have claimed that the MLP is efficient. I believe MD results are essential to support the claim for the high efficiency. It is ok if the author really don't want to present MD resuls in this manuscript if they state clearly that the MTP models are too computational expensive to do this.

Round 3

Reviewer 1 Report

Comments and Suggestions for Authors

I trust the authors and look forward to the promised future MD study of heat transport in the crystalline polymers. The current manuscript can be publised as is.